# Impact of lateral groundwater flow on hydrothermal conditions of the active layer in a high arctic hillslope setting

Alexandra Hamm[1,2] and Andrew Frampton[1,2]

[1]Department of Physical Geography, Stockholm University, Stockholm, Sweden
[2]Bolin Centre for Climate Research, Stockholm University, Stockholm, Sweden

**Correspondence:** Alexandra Hamm (alexandra.hamm@natgeo.su.se)

**Abstract.** Modeling the physical state of permafrost landscapes is a crucial addition to field observations in order to understand the feedback mechanisms between permafrost and the atmosphere within a warming climate. A common hypothesis in permafrost modeling is that vertical heat conduction is most relevant to derive subsurface temperatures. While this approach is mostly applicable to flat landscapes with little topography, landscapes with more topography are subject to lateral flow process as well. With our study, we contribute to the growing body of evidence that lateral surface- and subsurface processes can have a significant impact on permafrost temperatures and active layer properties. We use a numerical model to simulate two idealized hillslopes (a steep and a medium case) with inclinations that can be found in Adventdalen, Svalbard, and compare them to a flat control case. We find that ground temperatures within the active layer uphill are generally warmer than downhill in both slopes (with a difference of up to 0.8°C in the steep, and 0.6°C in the medium slope). Further, the slopes are found to be warmer in the uphill section and colder in the base of the slopes compared to the flat control case. As a result, maximum thaw depth increases by about 5 cm from the flat (0.98 m) to the medium (1.03 m) and the steep slope (1.03 m). Uphill warming on the slopes is explained by overall lower heat capacity, additional energy gain through infiltration, and lower evaporation rates due to drier conditions caused by subsurface runoff. The major governing process causing the cooling on the downslope side is heat loss to the atmosphere through evaporation in summer and enhanced heat loss in winter due to wetter conditions and resulting increased thermal conductivity. On a catchment scale, these results suggest that temperature distributions in sloped terrain can vary considerably compared to flat terrain, which might impact the response of subsurface hydrothermal conditions to ongoing climate change.

## 1 Introduction

Permafrost is defined as ground that remains below 0 °C for at least two consecutive years. It covers approximately 24% of the exposed land area in the northern hemisphere (Zhang et al., 1999) and stores about 1030 Pg of organic carbon in the upper 3 meters of soil (Hugelius et al., 2014). With increasing air temperatures in the Arctic, this carbon stock gets thawed out of the permafrost, exposing it to microbial decomposition and displacement. How much carbon gets released from the permafrost is strongly influenced by the depth of the active layer, the part of the soil that seasonally thaws out (e.g., Biskaborn et al., 2019). The correlation between increasing air temperature and depth of the active layer is well established (e.g., Zhang et al.,

1997; Isaksen et al., 2007; Frauenfeld et al., 2004). Especially high summer temperatures in dry environments have a direct impact on the development of the active layer in the same year (Isaksen et al., 2007). However, the effect of precipitation and hydrology in the active layer are less well understood as their effects are more dynamic and non-linear (e.g., Wen et al., 2014). Due to the low permeability of frozen ground, relevant hydrological processes are limited to the active layer. With increasing active layer thicknesses, more water can infiltrate into the ground and move laterally. The degradation of permafrost was found to decrease the seasonal variability of groundwater discharge into surface waters, changing the hydraulic connectivity in the subsurface and potentially also the solute transport capabilities (Frampton et al., 2011, 2013; Frampton and Destouni, 2015; Evans and Ge, 2017; McKenzie et al., 2021). Further, higher moisture abundance in the active layer regulates the decomposition of organic carbon (e.g., McGuire et al., 2009; Koven et al., 2011), can affect infrastructure built on the fragile frozen ground (e.g., de Grandpré et al., 2012), and can change the thermal properties of the permafrost (e.g., Schuh et al., 2017). Therefore, it is important to investigate the effect of hydrological and hydrogeological processes in permafrost landscapes.

In general, it is known that the amount of liquid water in the soil has a direct effect on its thermal properties (e.g., Iijima et al., 2010; Zhu et al., 2017). Wet soils are expected to conduct more heat towards the subsurface than dry soils in summer and, depending on the insulating effect of the snow cover, loose more energy to the atmosphere in winter (Kane et al., 2001). These conclusion are often based on 1D column representations of permafrost soils due to the assumption that vertical heat conduction is the major control of energy fluxes. For flat landscapes with little topography and low hydraulic gradients, these assumptions might be sufficient (Westermann et al., 2016; Langford et al., 2020). However, for permafrost underlying slopes, vertical conduction alone might not be able to explain permafrost degradation and seasonal active layer thaw. Due to the slopes and associated hydraulic gradients, lateral advection of water and energy might impact the ground thermal regime between up- and downhill locations. Especially in warmer, discontinuous permafrost landscapes, heat carried laterally by water has proven to be essential for subsurface temperatures and permafrost thaw (Sjöberg et al., 2016; Kurylyk et al., 2016; de Grandpré et al., 2012). This effect is even more enhanced and prolonged if water is gathering in water tracks on hillslopes (Evans et al., 2020). In a controlled laboratory experiment it was also found that subsurface flow can greatly enhance active layer thaw, but highly depends on the water temperature (Veuille et al., 2015). Further, groundwater flow along a hillslope in combination with preferential snow accumulation has shown how water and heat transport affect the emergence of a talik and how the talik can change the hydrological pathways within a permafrost hillslope (Jafarov et al., 2018). In high Arctic continuous permafrost landscapes, the effect of subsurface flow is expected to be less significant due to thin organic layers and generally low hydraulic conductivities (Loranty et al., 2018). In Yukon, Canada, it has been observed that vertical heat advection through snow-melt and summer rain infiltration on a road embankment change subsurface temperatures faster than through heat conduction from the surface (Chen et al., 2020).

Understanding and quantifying local-scale hydraulic permafrost processes helps to better constrain and inform global climate models and the feedback mechanisms between permafrost landscapes and the atmosphere, as permafrost is a key component of the climate system (Riseborough et al., 2008; Schuur et al., 2015). While field measurements are a vital source to achieve this, numerical modeling allows for applications with varying scenarios regarding environmental factors, such as climate setting

or slope inclination. Further, modeling can help untangle potential non-linear effects in the domain and dissect energy fluxes, which can be complex to measure in the field.

In this study, we investigate the role of hydrology on two idealized, 50 m long, high Arctic hillslopes and its effects on the active layer and ground temperatures, using a two-dimensional physically-based numerical model. We conducted a series of numerical model investigations representing typical hillslope environments and hydro-meteorological conditions of Adventdalen, Svalbard. The hillslopes are represented as idealized slopes with a steep (22°) and medium (11°) inclination and are compared to a reference case without inclination (flat case). We focus on absolute temperature differences between the uphill and downhill side in the slopes at several different depths within the active layer as well as the transect-wide active layer thickness in all cases. The model is controlled and driven by hydro-meteorological data and subsurface properties are consistent with site conditions. Our objectives are to understand and quantify the effects hillslope inclination have on active layer thermal and hydraulic dynamics of a permafrost catchment. Specifically, the following questions are investigated: (*i*) To what extent does hillslope inclination affect the ground temperatures in a permafrost catchment? (*ii*) To what extent is maximum active layer thickness and the volume of unfrozen soil affected by those differences? (*iii*) Which processes are responsible for the differences?

## 2  Data and method

The focus of this study is to investigate the effects that subsurface flow has on ground temperature and moisture in the active layer of a hillslope system located in a continuous permafrost environment. For this problem, the main governing processes which are relevant to consider are surface energy balances stemming from solar radiation, thermal insulation due to snow cover, sources of precipitation (snow, rain) with associated snow and/or ice accumulation, surface ponding and runoff on frozen or saturated ground, surface-subsurface infiltration in thawed and unsaturated ground, and subsurface water flow and heat transport in partially saturated, partially frozen ground. These processes are intricately coupled, in essence because water flow both above and below ground carries energy as a form of advective heat transport, and heat transport impacts the phase state of water, as liquid, ice or vapor, which in turn exerts control on water flow and heat conduction.

A numerical model is configured to correspond to site-specific conditions representative of hillslopes in Adventdalen, Svalbard, which is driven by atmospheric forcing and landscape data measured on-site. Even though site specific data was chosen to run the model, the aim of this paper is to provide a general idea of processes governing hydro-thermal responses of the active layer to groundwater flow while accounting for the full complexity of realistic boundary conditions. The model used is the Advanced Terrestrial Simulator (ATS v0.88, Coon et al., 2019). ATS is an open source, physically-based numerical model for coupled surface/subsurface thermal hydrology, specifically adopted for cold regions and permafrost applications (Painter et al., 2016).

A brief summary of the governing processes follows; for a full description see the cited references. ATS solves coupled conservation equations for energy and water mass transport, considering both above and below ground processes, based on a multiphysics framework (Painter, 2011; Coon et al., 2016). The available energy at the surface-subsurface interface drives

subsurface heat transport, and is obtained by solving for a surface energy balance equation (Atchley et al., 2015). Snow and ice on the surface affect heat conduction by reducing or increasing thermal conductivity, and subsequently impact heat transfer to the subsurface. Snow and ice are also subject to melting and ponding and can provide a source of water infiltration and/or surface runoff. Unfrozen water flow on the surface follows the Manning equation (Painter et al., 2016).

In the subsurface, conductive heat transport follows Fourier's law, with an effective thermal conductivity based on the material properties and accounting for the phase state of the pore-filling fluid (as ice, liquid or air) (Painter, 2011). Advective heat transport occurs as heat carried by water movement in the porous media. Subsurface flow of water is governed by the extended Darcy law for partially saturated flow, where phase transitions follow the Clausius-Clapeyron relationship accounting for latent heat transfer. Soil moisture retention curves, adopting a van Genuchten formulation, are used to describe effective permeability in the variably saturated pore space, accounting for the presence of air and ice, where ice is considered an immobile phase, causing a reduction in available porosity (Painter and Karra, 2014). Volume change for the phase changes between liquid and ice is accounted for by a pore compressible factor. Furthermore, ATS adopts a flux-conserving finite volume solution scheme and supports unstructured meshes, thus can conveniently be used for applications in 1D, 2D and 3D, accounting for vertical and lateral processes in all dimensions considered.

## 2.1 Field data

Svalbard is located at 78°N and 15°E and therefore represents high-Arctic climate. Active layer thickness in Adventdalen has increased with a rate of $0.7\,\mathrm{cm\,yr^{-1}}$ over the last decades and currently ranges between 0.9 and 1.1 m (Strand et al., 2020). The observational weather data to drive the model (hereinafter referred to as the forcing dataset) is derived from an automatic weather station located in Adventdalen (78.2°N 15.87°E) operated by the University Center in Svalbard, which measures air temperature, incoming short- and longwave radiation, relative humidity, and wind speed.

Precipitation measurements are retrieved from the long-term weather station at Longyearbyen airport (9 km west of the Adventdalen weather station; 78.24°N 15.51°E) operated by the Norwegian Meteorological Institute. Precipitation is retrieved as daily values representing daily cumulative rain- or snowfall. Air temperature, relative humidity, and wind speed are measured in one-second intervals, radiation in five-minute intervals, and represent instantaneous values. The time period of measurements used in this study is 2013 to 2019 and measurements are aggregated into daily sums or averages.

To create the forcing dataset, mean values of each variable for every day of the year (day-of-year average) between 2013 and 2019 are calculated to obtain a representation of current average weather conditions. Further data processing involves the classification of precipitation as rain if mean daily air temperatures are above 0° C, and as snow if air temperatures are below 0° C. An adjustment for precipitation undercatch in Svalbard has been suggested to be 1.85 for snow and 1.15 for rain (Førland and Hanssen-Bauer, 2000), and therefore precipitation is multiplied by these respective factors. This results in an average annual sum of 330 mm for the period 2013–2019. The annual sums of rain (160 mm) and snow (170 mm w.e.) are then redistributed to equal daily amounts during the rain- and snow period, respectively. The mean annual air temperature for the calculated averages over this time period is -2.8° C.

**Table 1.** Physical properties of the subsurface material. Notations $S_{uf}$ and $D_{uf}$ denote saturated, unfrozen and dry, unfrozen conditions.

| Material Property | Unit | Value |
|---|---|---|
| Porosity | $m^3\,m^{-3}$ | 0.4 |
| Permeability | $m^2$ | $2 \times 10^{-13}$ |
| Density | $kg\,m^{-3}$ | 2650 |
| Van Genuchten $\alpha$ | $Pa^{-1}$ | $8 \times 10^{-4}$ |
| Van Genuchten $m$ | - | 0.2 |
| Thermal conductivity $S_{uf}$ | $W\,m^{-1}\,K^{-1}$ | 1.7 |
| Thermal conductivity $D_{uf}$ | $W\,m^{-1}\,K^{-1}$ | 0.27 |
| Specific heat capacity | $J\,kg^{-1}\,K^{-1}$ | 850 |

Thereby, the resulting forcing data set consists of daily values based on the average for each day of the year between 2013 and 2019 for wind speed, air temperature, incoming shortwave radiation, relative humidity, incoming longwave radiation, rain precipitation, and snow precipitation (Fig. S1). This yearly cycle of average weather data is then repeated 100 times (corresponding to 100 annual cycles) to create the forcing dataset needed to initialize and run the simulations, as described in Section 2.2.2.

**2.2   Simulation configurations**

Three idealized model cases are considered; a steep case with a 22° slope, a medium case with a 11° slope, and a flat case with a 0° slope. The flat case is used primarily as reference to evaluate effects of inclination and to normalize quantities for analysis. The model cases are identical in all respects other than inclination. Note that the elevation difference between the uppermost and lowermost part of the slopes is 10 and 20 m for the medium and steep slope, respectively, but temperature does not change

depending on altitude in this setup.

The inclinations are based on slopes as they can be found in Adventdalen and its southern tributaries mostly below 200 m elevation. Geologically, the slopes are located within the Carolinefjellet formation, which mainly consists of shale, siltstone and sandstone (Norwegian polar institute). All hillslope areas greater than 5° inclination in the area of question are shown in Fig. 1a and b. An aerial image of the area is shown in panel c. In the same way as in panel a and b, slopes have been calculated

in regions around the Arctic to evaluate how representative the slopes considered in this study are for the Arctic as a whole (Fig. 1d). It can be seen that even though great parts of the landscape are rather flat ($< 5°$: 40–84%), all regions also have slopes in both categories (5–15°: 12–30% and 15–25°: 2–14%) or even steeper ($>25°$: 1–19%). For information on the methodology used to derive information about slope inclinations around the Arctic and for the values of the pie charts please see section 1 and Table S1 in the supplementary material.

The flat control case corresponds to areas with no considerable inclination as they can be found in the Adventdalen valley bottom. These areas are characterized by holocene glaci-fluvial deposits (Norwegian polar institute). It can be seen that some

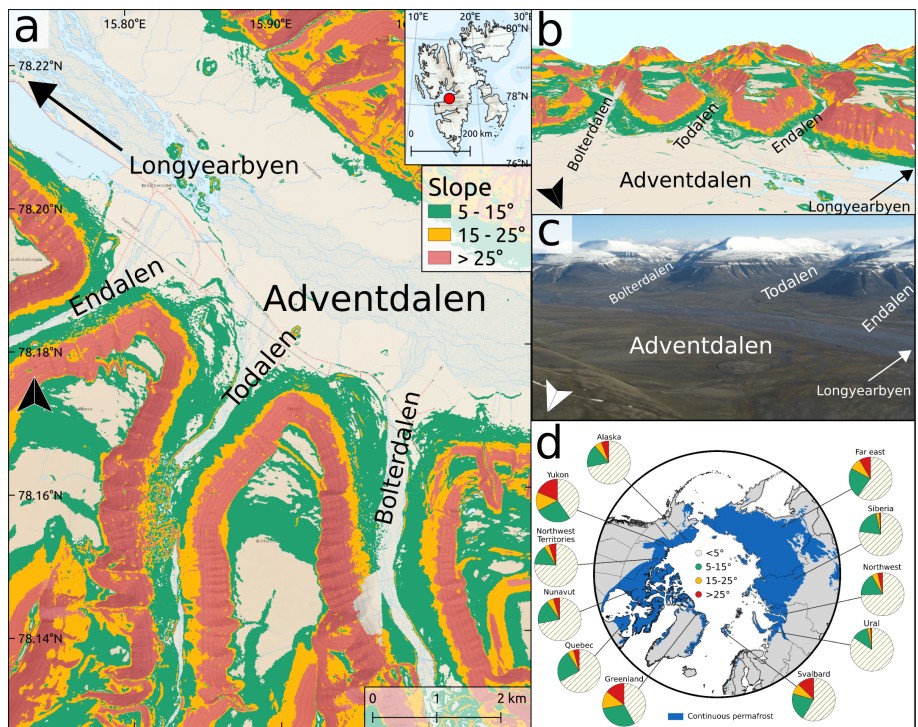

**Figure 1.** Overview over the general study area. **a** shows a map with slope inclinations greater than $5°$ along some of Adventdalen's southern tributaries (Endalen, Todalen, Bolterdalen). **b**: 3D view of the valleys and slope of the map in panel a. Basemap data has been retrieved from the Norwegian polar institute. Inclination values are based on elevations from the Arctic DEM (10 m resolution; Porter et al., 2018). **c** shows an aerial image of Adventdalen overlooking the same area as in the maps in panels a and b. The picture was taken from a helicopter by A. Skosgslund (Norwegian polar institute). **d**: Overview over slopes in Arctic continuous permafrost regions based on different administrative areas following the classification in panels a and b.

slopes end in the tributaries of Adventdalen (Endalen, Todalen, Bolterdalen), which contain seasonal river systems. Other, mainly north facing, slopes do not necessarily end in a surface water body but somewhere in the flat part of the Adventdalen valley bottom. This is important for the choice of boundary conditions in the model domain, which regulates the water flux out of the domain. Potential boundary conditions for this set-up could be either a closed boundary (no outflow), an open boundary (outflow through the surface and subsurface), or a constant head boundary, which would indicate a persistent river and allow for groundwater discharge into the river.

### 2.2.1 Model domain and boundary conditions

To represent the slopes in ATS, each case has its own mesh. The slope-meshes consist of a sloped part ($x = 0$–50 m) with a constant slope of 11 and $22°$, and an adjacent flat valley bottom ($x = 50$–66 m). Each model case has a corresponding surface and subsurface mesh. The surface mesh is a 2D layer which extends 66 m in $x$-direction and 1 m in $y$-direction,

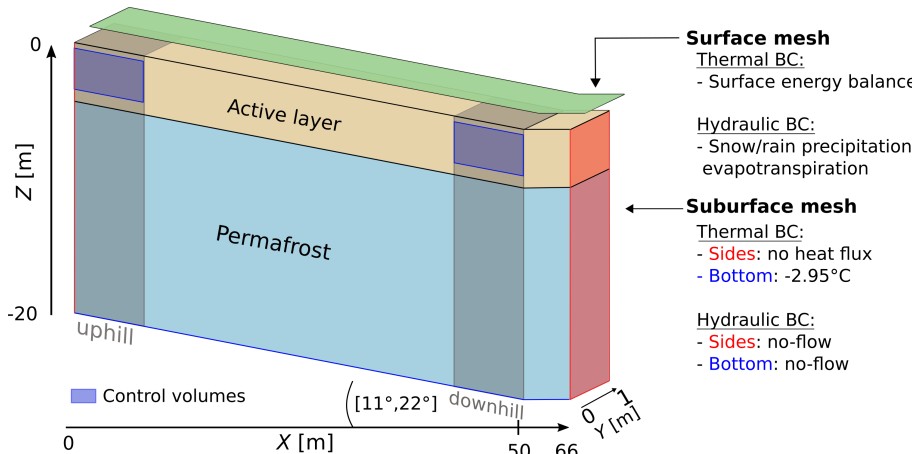

**Figure 2.** Conceptual representation of the surface and subsurface modeling domain. Grey shaded areas on either side of the transect indicate the uphill and downhill observation locations, red indicates the sides of the model, blue boxes represent the control volumes (CV) and a blue line at the bottom indicates the bottom boundary. Thermal-hydraulic boundary conditions (BC) on the surface, sides and bottom are listed on the right.

and the subsurface mesh extends 66 m in $x$-direction, 1 m in $y$-direction and 20 m in the $z$-direction (Fig. 2). Both have a lateral resolution of 2 m yielding 33 mesh elements along the $x$-direction. Only one element with unit width is assigned in the transverse $y$-direction. Thus, the subsurface elements are 3D volumes and yield volumetric flow quantities, but the model
setup effectively represents a 2D transect of the surface-subsurface system with unit width (for actual mesh representations please refer to Fig. S2 in the supplementary material). This approach has been found to be a valid simplification of complex slope systems (Jafarov et al., 2018; Jan and Painter, 2020). In the uppermost meter of each column, cells are generated with a higher resolution of 0.05 m height in the vertical direction, in order to improve the spatial resolution of the active layer. With increasing depth, cell thickness gradually increases (up to max. ∼1.5 m cell thickness).
All cases assume a homogeneous material throughout the model domain consistent with mineral soils typically encountered in the area (Schuh et al., 2017). We do not consider an organic layer in our setup as they are absent on most parts of the slopes in the Adventdalen area. The physical and material properties used to describe the subsurface domain (Table 1) are consistent with a previous study based on the UNISCALM site in Adventdalen (78.2°N 15.75°E), which showed good agreement with subsurface measurements and produced realistic active layer depths (Schuh et al., 2017).
The boundary conditions for the subsurface domain are prescribed as no-flow boundaries on the left and right side, and at the bottom. Therefore, the uphill end conceptually represents a water divide, as no flow enters the domain from further up. The downhill end of the transect represents the valley bottom, and allows for water accumulation and potential ponding on the surface. The right-most boundary reflects a symmetry boundary, representing a simplified U-shaped valley bottom. This valley bottom ($x = 50$–66 m) is needed to avoid edge/boundary effects and is omitted in the analysis of the results. The domain size is
chosen to represent a generic hillslope that provides a reasonable trade off between model resolution and computational effort.

The bottom temperature is set to -2.95° C, which has been found to be the temperature at 19 m depth in a borehole in Endalen, one of Adventdalen's tributaries (Hanssen-Bauer et al., 2018). The borehole is located on a slope and therefore assumed to be representative for other slopes in Adventdalen. As the borehole temperature experiences a linear increasing trend throughout the 2013–2019 period, the mean value of the same period is used. The surface is subject to hydro-meteorological conditions measured on-site (the forcing dataset), which effectively drives the dynamics of heat and water flow through the model system. Precipitation is added as snow and rain on the surface, which allows for infiltration, and heat is supplied by the surface energy balance. Water can leave the system via evaporation and the surface allows for snow and ice accumulation as well as water ponding. Snow distribution for these simulations has intentionally been disabled, in order to yield the same snow accumulation on the surface of the model domain. This is due to the fact that an accumulation of all available snow on the downhill side of the slope is not realistic and the fact that this would considerably increase the complexity of our analysis and the disentanglement of the effects of groundwater flow on the hydrothermal state of the active layer, which is the focus of this study.

The model output is given as cell values in selected cells of the sloped part of the model domain. Analysis of these values includes temperature, saturation thermal conductivity, and heat capacity, extracted at 0.1 m, 0.2 m, 0.5 m and 1 m depth at an uphill and downhill location of each model domain. These depths are chosen as they represent the near-surface soil conditions, the middle of the active layer and the bottom of the active layer.

For analysis of fluxes, two control volumes (CV) are defined, also located uphill and downhill (see Fig. 2). The uphill CV extends from 0 to 2 m in $x$-direction, 0 to 1 m in $y$-direction, and from -0.1 to -0.6 m in $z$-direction. The volume of the box is thus approximately ~1 m$^3$. The downhill CV is defined as a box from 48 to 50 m in $x$-direction, 0 to 1 m in $y$-direction, and -0.1 to -0.6 m in $z$-direction (~1 m$^3$). The upper boundary is moved 0.1 m below the surface, as the surface itself includes more processes than subsurface faces, which would complicate the comparison to the bottom-boundary face of the CV. Each face of the box is used to capture advected and diffusive energy flux, and mass flux into and out of the domain during the simulation. Lateral fluxes in the CVs are only represented on the right boundary of the uphill CV (flux directed outward) and on the left boundary of the downhill CV (flux directed inward). We placed the CVs at these locations to capture the most extreme values within the sloped part of the domain and to link them to the cell values in the same locations.

### 2.2.2 Model initialization and spin-up

Model initialization and spin-up is conducted with a three-step procedure following previously established routines for permafrost-hydrological modeling (Karra et al., 2014; Painter et al., 2016; Pannetier and Frampton, 2016; Jafarov et al., 2018; Jan and Painter, 2020; Sjöberg et al., 2021). First, a single 1D column is used to establish hydrostatic conditions with the water table at a target depth, using pressure boundary conditions for the top and bottom faces of the model. Second, the soil and water in the column is cooled from below with an assigned sub-zero bottom temperature, until the column is fully frozen and reaches a cryotic steady-state. In the third step, the forcing dataset (Section 2.1) is used to bring the thermal-hydraulic conditions of the column model into an annual steady state. The annual steady state is achieved by repeating the forcing data set for 50 annual cycles, corresponding to 50 years of simulation, after which inter-annual temperature differences throughout the column are

less than 0.01°C. This procedure is necessary to obtain a physically consistent system which can be used as initial condition for the main simulation runs.

In the (final) fourth step, the resulting state from the 1D single column spin-up model is mapped to each of the 33 columns of the hillslope transect model. Thereafter, the same forcing dataset (Section 2.1) is used again to run the simulations, now in the full domain allowing for all lateral and vertical dynamic processes to occur. The full model is run for 100 annual cycles, corresponding to 100 years of simulation. The first 99 years are considered as spin-up, to obtain an annually periodic steady-state for the entire surface-subsurface hillslope system in the 2D model domain. The final year of the simulation (year 99 to year 100) is then considered as the simulation result, used for analysis in this study. Thus, it is equivalent to a representation of the hydrothermal state of the subsurface corresponding to the current 2013–2019 average weather conditions. The initialization procedure is repeated for each model case considered, to ensure effects of hillslope inclination and wetness conditions are embedded in the final simulation results.

## 3   Results and discussion

### 3.1   Temporal analysis of ground temperatures

Daily ground temperatures in the active layer (0.1 m, 0.2 m and 0.5 m depth) and near the permafrost table (1 m depth) vary between the different inclination cases, and there are also temperature differences between the uphill and downhill observation locations. Additionally, timing of thaw and freeze-up varies between cases. To enable a systematic study of the impact of the different hillslope inclinations, we consider daily temperature differences $\Delta T_I$ between the steep slope and flat case (steep-flat), as well as between the medium slope and flat case (medium-flat). We also consider daily temperature differences $\Delta T_E$ between uphill and downhill observation points (uphill-downhill), corresponding to different elevations along a hillslope (Fig. 3). A time series of daily subsurface temperatures in each depth and location can be found in Fig. S3 in the supplementary information.

There is variability in these temperature differences over the year, with most pronounced differences occurring during the warm season, typically including a peak just after the thaw period and another peak after freeze-up, indicating greatest differences occurring during these times. Between the uphill and downhill side in the steep and medium slope (Figures 3a,b), it can be seen that the uphill side is warmer than the downhill side throughout the year (positive $\Delta T_E$), with two short exceptions just after thaw and after freeze-up (negative $\Delta T_E$). The warming is strongest in summer and occurs first close to the surface (0.1 m, orange line) exhibiting a temporal lag effect with depth. At 1 m depth (yellow) the warming effect is delayed and smaller due to the overall colder temperatures near the permafrost table. Just after freeze-up, however, differences at 1 m depth are largest as cooling close to the permafrost table occurs faster than the the rest of the active layer due to the presence of the permafrost. The overall greatest temperature differences can be seen in the middle of the active layer (0.5 m depth) around July in the steep case (∼0.8°C warmer than downhill).

Temperature differences $\Delta T_I$ between the steep slope and the flat case (Fig. 3c,d) and between the medium slope and the flat case (Fig. 3e,f) show that on the uphill side, the slopes are warmer in summer, colder after freeze-up and very similar to the

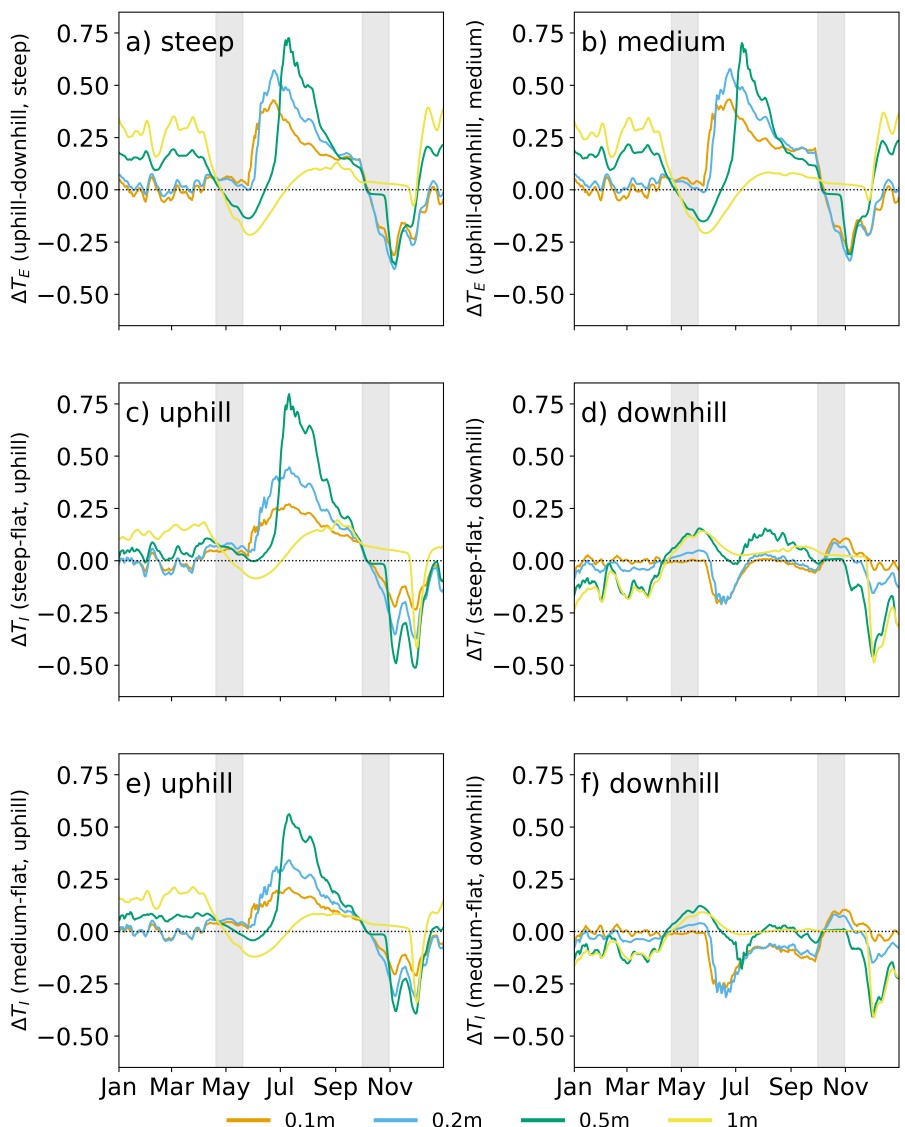

**Figure 3.** Daily temperature differences (averaged over a 7-day window) $\Delta T_E$ **(a, b)** and $\Delta T_I$ **(c, d, e, f)** in four different depths within the active layer. Grey shaded areas indicate periods of thaw and freeze-up. Temperature differences (in °C) between locations ($\Delta T_E$) are calculated by subtracting the downhill temperature from the uphill temperature. Resulting positive values indicate warmer temperatures uphill, while negative values indicate colder temperatures. Temperature differences between slopes ($\Delta T_I$) are calculated by subtracting the flat case temperatures from each sloped case (steep and medium). Positive values indicate that the slopes are warmer, while negative values imply that the slopes are colder compared to the flat case. Due to the unsmoothed forcing data and downwards propagation of the surface signal, day-to-day changes can be considerable.

**Table 2.** Average temperature of the entire transect up to 1.2 m depth for each day of the snapshots.

| | average temperature [°C] | | | | | |
| --- | --- | --- | --- | --- | --- | --- |
| | Jun 30 | Jul 20 | Aug 9 | Oct 28 | Nov 27 | Dec 7 |
| steep | 0.73 | 1.72 | 2.02 | -0.7 | -0.85 | -5.14 |
| medium | 0.65 | 1.6 | 1.9 | -0.68 | -0.82 | -5.1 |
| flat | 0.56 | 1.47 | 1.79 | -0.64 | -0.78 | -5.06 |

flat case in winter. On the downhill side, the slopes are colder than the flat case in winter, after thaw and after freeze-up. This is especially true for the medium slope and near-surface temperatures. Deeper layers have similar temperatures to the flat case or are even warmer. An overview of the yearly maximum temperature differences $\Delta T_I$ and $\Delta T_E$ is given in Tables S2 and S3 in the supplementary material.

### 3.2 Spatial analysis of ground temperatures

The greatest temperature difference along the subsurface transect occurs between the two outermost slope locations (at $x=0$ m and $x=50$ m), corresponding to the two locations farthest apart along the hillslope. To better visualize the ground temperature differences between cases throughout the subsurface domain, the temperature difference between the steep and the flat case (Fig 4a), and the medium and flat case (Fig 4b) in the upper 1.2 m of the subsurface are considered. Note that Fig 4 shows cell-based temperature differences between cases; thus slope inclination is not depicted. The upper three plots in each panel (a and b) show snapshots of temperature differences during thaw (June) and summer (July, August), and the lower three plots show temperature differences during freeze-up (October, November) and winter (December). In both cases, the dates are separated by 20 days. For each day, the 0,°C isotherm(s) from the steep and medium case respectively is (are) represented as black dotted line(s). During thaw they represent the maximum depth at which temperatures exceed 0 °C (i.e. the soil above is unfrozen, the soil below is frozen) and during freeze-up, they show unfrozen parts of the subsurface (i.e. the soil between the lines is unfrozen, the soil outside is frozen). The average temperature in this volume of the subsurface (upper 1.2 m) is given in Table 2.

Ground temperatures in the sloped cases are generally warmer than in the flat case during thaw and summer (red shades). The temperature differences are greatest near the progressing thaw front, i.e. near the 0 °C isotherm, as well as on the uphill side ($x=0$ m), but a gradual change towards similar temperatures as the flat case (red to white) can be observed in the lateral direction (increasing $x$). The temperatures below the permafrost table (at approximately 1 m depth) are only slightly warmer in the steep case, and essentially unchanged in the medium case, for the summer snapshots.

During freeze-up (October 28 and November 17 snapshots) the sloped cases are generally colder in the topsoil and warmer in the permafrost compared to the flat case. By winter (December 7) almost the entire subsurface (up to 1.2 m) of the steep slope becomes colder than the flat case (light blue). Only some areas on the downhill half of the transect remain slightly warmer (yet below freezing) than the flat case, while the very last column of the slope is significantly colder (dark blue). The patterns

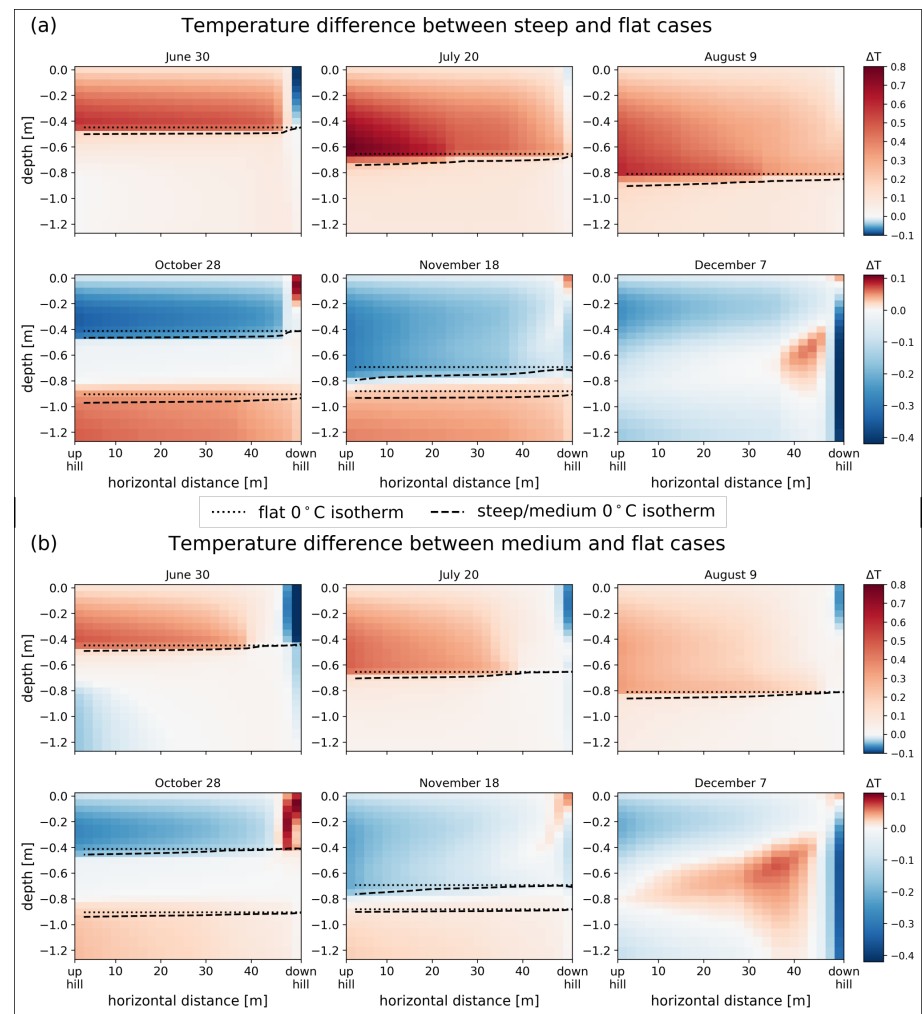

**Figure 4.** Temperature difference between **a** the steep and the flat case and **b** the medium and flat case at six selected dates highlighting thaw, summer, freeze-up and winter. Red colors indicate warmer temperatures in the hillslope cases than in the flat case, blue colors indicate cooler temperatures (note the color scale differs between summer and winter comparisons). The black dashed lines indicate the $0\,°\mathrm{C}$ isotherm(s) in the corresponding hillslope cases (steep and medium) at the respective dates. The $0\,°\mathrm{C}$ isotherm lines of the flat case are represented by dotted lines. During freeze-up, it can be seen that two-sided freezing occurs. (For clarity, only the upper $1.2\,\mathrm{m}$ of the simulation domain is shown).

seen in both December 7 plots (red patches between -0.2 and -1.2 m) are consequences of the timing of freezing in the slopes. While the flat case freezes uniformly, the active layer in the slopes freezes faster uphill and slower downhill, causing those

temperature differences.

The $0\,°\mathrm{C}$ isotherms show that on July 20, the steep slope develops a deeper thawing front in the first $20\,\mathrm{m}$ of the transect compared to the flat slope. On August 9, the steep slope is warmer throughout most of the active layer thickness (approx. 1 m)

of the transect; only the last column shows similar near-surface temperatures as in the flat case. In the medium slope case, the first 35 to 40 m ($x$=0–35/40 m) of the slope show warmer temperatures and deeper progressing thaw fronts on all dates, compared to the rest of the slope. The last 10 to 15 m ($x$=35/40–50 m) exhibit shallower thaw fronts. In both, the October and November snapshots, the freeze-up process shows how the transects freezes from top to bottom, as well as slowly from the permafrost table upwards, thereby exhibiting two-sided freezing. Between October 28 and November 18 it can therefore be seen that even though the ground appears to be frozen from the surface, it is still unfrozen in the lower part of the active layer. By December 7, the entire active layer is frozen.

The spatial mean active layer depth in the steep slope on the date of maximum active layer depth is 1.03 m (min.:1.03 m, max.: 1.03 m along the transect). The medium slope exhibits a smaller uphill warming than the steep slope resulting in a spatial mean active layer depth on the date of maximum active layer depth of 0.986 m (min.:0.975 m, max.:1.030 m along the transect), which is only slightly deeper than in the flat case (0.975 m). In general, these thaw depths are in line with average active layer thicknesses measured in Adventdalen (0.9 to 1.1 m; Schuh et al. (2017); Strand et al. (2020)). Daily values for thaw depth in each case can be found in Fig. S4 in the supplementary information.

Overall, we observe that the steep slope case has a notable influence on thaw propagation and active layer thickness, which we attribute to an increase in ground temperatures compared to the flat case, observed primarily in the center-uphill side of the subsurface during most of the summer period. The medium sloped case only shows a marginal increase in maximum thaw depth, but it can be seen that both slopes start thawing earlier and the day of maximum thaw depth is reached earlier compared to the flat case, while freeze-up is delayed. Thaw begins on May 22 in the steep case and May 24 in the medium and flat case. Freeze-up is complete on November 23 in the steep case and on November 22 in the medium and flat case. This can also be seen by integrating the total volume of unfrozen soil over the warm season (defined as days with at least one unfrozen cell in the subsurface model domain, here resulting in May 15 to October 2; 140 days). The steep slope amounts to a total volume of 2936 m$^3$ (or an average of 20.97 m$^3$ per day), the medium slope amounts to 2905 m$^3$ (average 20.75 m$^3$ per day), and the flat slope amounts to 2885 m$^3$ (average of 20.61 m$^3$ per day). This indicates that the slopes in general have a greater unfrozen volume of soil, even though active layer depth in the medium case is not substantially different. Hence, the warming effect due to slope inclination not only plays a role in the vertical soil profile, but also in the timing of freeze and thaw.

### 3.3 Saturation, thermal conductivity and heat capacity

Due to gravitational flow of water during the warm period, moisture is drained from the uphill side and accumulates on the downhill side, reducing liquid saturation uphill and increasing it downhill when compared against the flat reference case, which is not subject to lateral flow (Fig. 5, first column). This leads to differences in ice saturation during the frozen period (Fig. 5, second column), specifically reduced ice saturation uphill and increased downhill. Consequently, the uphill side of the sloped cases experience increased air saturation (Fig. 5, third column), which yields a considerably lower effective thermal conductivity during winter and slightly lower effective thermal conductivity during summer (Fig. 5, fourth column). Similarly, the downhill side has reduced air saturation (Fig. 5, third column), yielding greater effective thermal conductivity; considerably

greater during winter and slightly greater during summer (Fig. 5, fourth column). A 2D representation of liquid-, ice-, and air saturation analogous to Fig. 4 can be found in Fig. S5 in the supplementary information.

Considering the little snow cover in winter (max. 0.01 m, see Fig. S6), the effect of differences in thermal conductivity should be an enhanced heat loss (cooling of the ground) during winter, and slightly enhanced heat gain (warming of ground) during summer, when compared against the flat reference case. Furthermore, differences in liquid saturation change the bulk heat capacity (Fig. 5 fifth column) of the two sections. While it is reduced in the uphill section, it is higher in the downhill section of the domain. This causes the uphill section to warm up and cool down faster than the downhill section and contribute to overall warmer summer temperatures uphill. Downhill, it slows the warming and cooling process down. A 2D representation of differences in heat capacity between the steep and flat and the medium and flat case throughout the upper 1.2 m of the domain can be found in Fig. S7 in the supplementary information (analogous to Fig. 4).

Recall the previous discussion on temperature differences between the sloped and flat cases (Section 3.2). The uphill sides of the sloped domains (Fig. 3c,e) are slightly drier at depths 0.2 m, 0.5 m and 1 m, both for summer with less liquid saturation, and winter with less ice saturation (Fig. 5, first and second columns, respectively). This slightly reduces effective thermal conductivity with respect to the flat case at those depths, mainly in winter and slightly discernible also in summer (Fig. 5, fourth column). Thus, when compared against the flat reference case, the uphill side of the inclined cases should exhibit warmer ground temperatures during winter due to reduced thermal conductivity (greater insulation) and hence reduced heat loss. During summer, the reduced thermal conductivity is only minor, but if anything may lead to a reduced heat gain, leading to slightly cooler ground when compared to the flat case. However, this is not entirely consistent with the previously observed temperature differences for the uphill side (see Figures 3c,e). While winter temperature differences are positive (after the freeze-up effect) and hence are consistent with smaller heat loss to the atmosphere than in the flat case, summer temperatures are warmer in the sloped cases, not cooler. This can partially be explained by reduced heat capacity in the uphill section, which allows for faster warming and overall higher temperatures. This effect potentially outweighs the reduced heat conduction from the atmosphere into the ground through lower thermal conductivity, but does not explain the entire difference.

Next, consider the downhill side (Fig. 3d,f). The sloped cases experience cooler winter temperatures, especially shortly after freeze-up in late November. This is consistent with the differences in effective thermal conductivity, as an increased thermal conductivity during cold periods enables an enhanced ground heat loss, yielding cooler winter ground temperatures. However, summer temperatures exhibit very similar or even cooler temperatures than the flat case (Fig. 3d, and especially in f. This is not consistent with the increased effective thermal conductivity summertime, as it should enhance heat uptake to the ground, leading to warmer ground temperatures. Thus, we conclude changes in effective thermal conductivity alone do not suffice to explain the negative temperature differences on the downhill side for the two hillslope cases in comparison to the flat case. Considering heat capacity, however, it can be expected that wetter soils in the downhill section require more heat to warm up and thus remain slightly colder, which can counteract the effect of thermal conductivity to the findings in Fig. 3d,f.

When only comparing the two observation locations uphill vs. downhill within the slopes (Fig. 3a,b), similar effects as previously described can be seen. Again, winter differences can be explained by increased heat loss to the atmosphere due to greater thermal conductivity on the downhill side. Summer differences cannot be explained by changes in saturation and

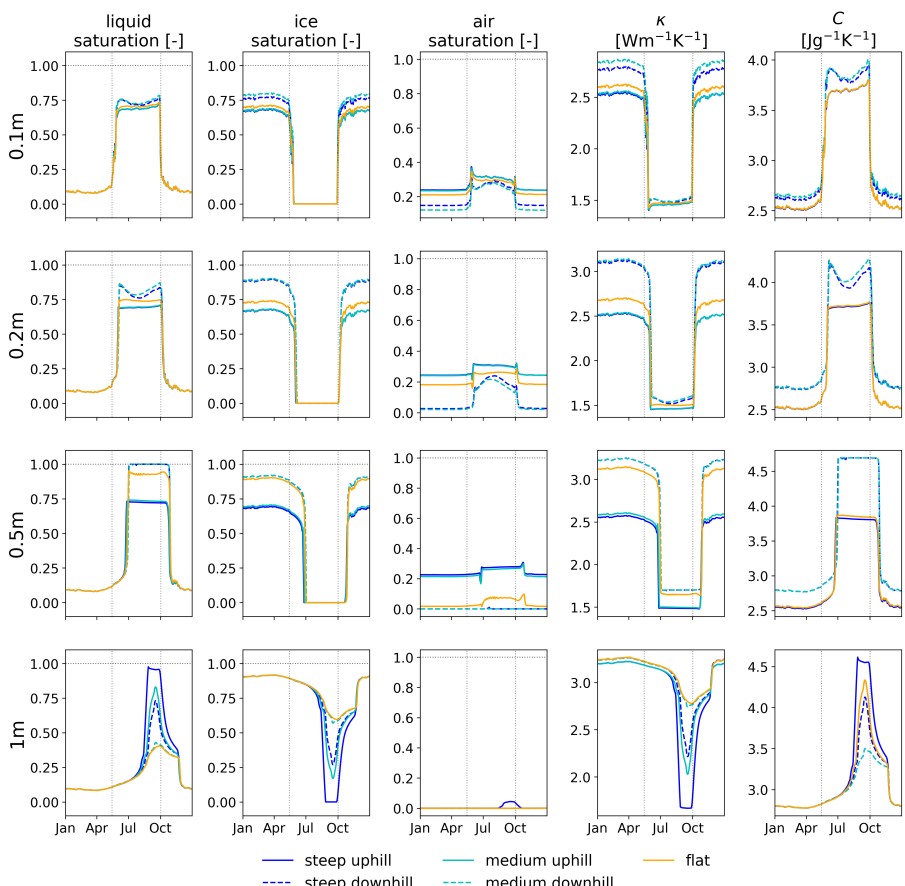

**Figure 5.** Daily values for liquid, ice, and air saturation (columns 1–3), thermal conductivity ($\kappa$; column 4) and heat capacity ($C$; column 5) at 0.1, 0.2, 0.5 and 1 m depth (rows 1–4). Colors represent the three different cases and solid and dashed lines mark uphill and downhill sides, respectively. The horizontal dashed lines in the saturation plots indicate 100% saturation. The vertical dashed lines mark the first and last day at which ground surface temperatures exceed 0°C.

effective thermal conductivity alone, but are partly attributable to lower heat capacity. However, these described effects are not sufficient to explain the full range of temperature difference.

In summary, moisture redistribution mainly causes differences in thermal conductivity and heat capacity between the uphill and downhill sections (Fig. 6). Thermal conductivity mainly affects energy transport by conduction, and heat capacity atten-
345 uates transport by storage. However, to fully understand the effects of energy transport on ground temperatures, a complete analysis of energy fluxes is needed, which is discussed in the next section.

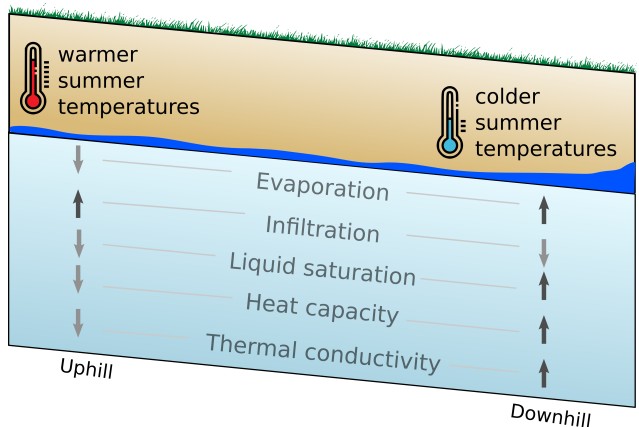

**Figure 6.** Conceptual diagram of the effects of saturation on ground temperatures in the active layer in summer time. The arrows indicate if the quantity is increased (up, dark) or decreased (down, light).

## 3.4 Energy fluxes

Vertical and lateral energy fluxes are calculated through the faces of two control volumes (CV) in the subsurface domains; one placed on the uphill side and the other on the downhill side (see Fig. 2). The objective is to investigate fluxes within the active
layer, hence the CVs extend from -0.1 m depth to -0.6 m depth below the surface. Daily flux values averaged over a 90-day window are considered, defined as positive if entering the CV, and negative if leaving the CV. Diffusive heat flux (energy transport by conduction) and advective heat flux (energy transport by water flow) obtained this way are shown in Figs. 7 and 8, respectively, for both the uphill CV (solid lines) and downhill CV (dashed lines). The central box (conceptually) aids the interpretation of the fluxes across corresponding faces of the control volume. Fluxes across the top and bottom faces represent
fluxes at $z$=0.1 and $z$=0.6 m depth, while fluxes across the left and right faces represent fluxes across vertical faces at $x$=48 and $x$=2 m, respectively. The distance is given as distance from the left domain boundary ($x$=0 m). Note the lateral fluxes are only displayed on one of the vertical faces of the CVs as the opposing sides ($x$=50 and $x$=0 m) represent the edges of the slope. Fluxes can vary by more than one order of magnitude between cases, which results in different ranges of values for the vertical axes. Daily ratios between advective and diffusive energy fluxes (Péclet number) for all faces of the CVs are given in
the supplementary information (Fig. S8)

The most pronounced flux is vertical heat diffusion near the surface (-20–20 W m$^{-2}$), which shows little relative difference between the hillslope cases. Across the top face, i.e. at 0.1 m depth, the downhill CVs (Fig. 7a, dashed) show slightly greater heat gain through heat diffusion in summer (up to ∼2.5 W m$^{-2}$) and slightly greater heat loss during freeze-up, than the uphill CVs (solid). Winter diffusive heat fluxes are almost identical. Lateral heat diffusion is smaller, but more pronounced and quite
variable in the downhill CVs (Fig. 7c, dashed, -0.01–0.15 W m$^{-2}$). It is highest just before freeze-up and in winter, which is attributable to a high temperature gradient between the penultimate and the last column in the slope domain.

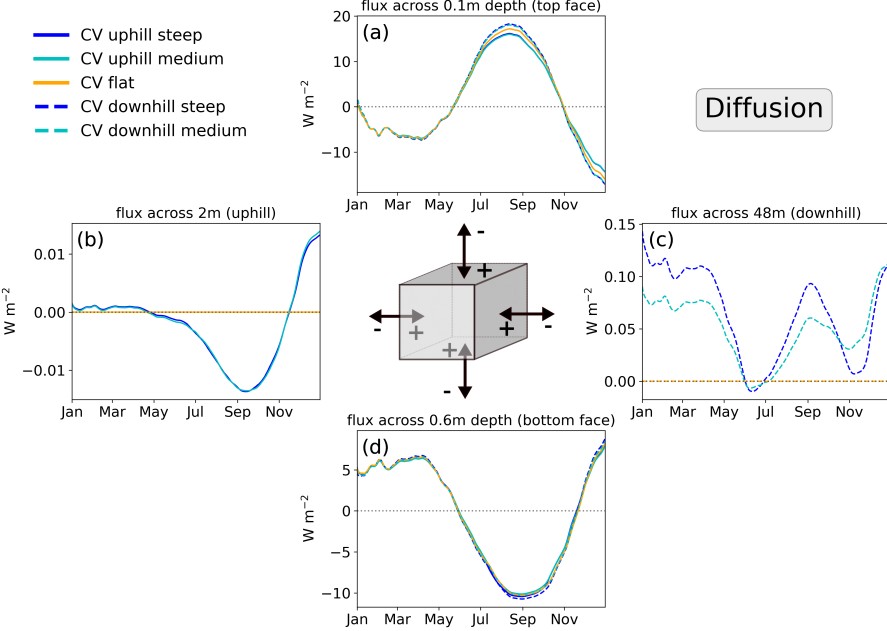

**Figure 7.** Daily values of diffusive heat flux on the faces of the control volume (CV; 90-day moving average) at the uphill (solid) and downhill (dashed) CV locations. Colors represent the steep (blue), medium (cyan) and flat (yellow) case, respectively. The sign convention adopted is positive values represent heat entering the CV and negative values leaving the CV. Due to the definition of the CV boundaries, lateral fluxes only occur on the right face for CV up and on the left side for CV down.

In the uphill CV (Fig. 7b, solid), the lateral heat diffusion is more than one order of magnitude smaller than in the downhill CV (-0.01–0.015 $\mathrm{W\,m^{-2}}$) and heat is being lost in summer, but gained after freeze-up in winter. This is also consistent with the warming and the reduced effective thermal conductivity observed on the uphill side of the domain, which combined should yield a decreased heat flux.

Advective heat flux magnitudes are generally much smaller than diffusive flux magnitudes (Fig. 8). Note that advective fluxes only occur in summer and during freeze-up, i.e. when unfrozen water is available for flow, and further only occur in lateral direction for the sloped cases (steep and medium); the flat case exhibits zero values for advective (lateral) flux, as expected. Note also that the magnitude of lateral advective heat flux is about one order of magnitude larger on the downhill side (Fig. 8c, dashed) than on the uphill side (Fig. 8b, solid). As water flows and accumulates downhill, the heat carried by water causes the lateral heat flux magnitude to increases downhill. This can be seen by the flux magnitude across the $x$=2 m face (Fig. 8b, uphill) being much smaller than across the $x$=48 m face (Fig. 8c, downhill). Thus, the increase in lateral advective heat flux should contribute to warmer ground temperatures on the downslope side of the domain. However, summer temperature differences between the up- and downhill column show that the downhill columns ($x$=48–50 m) are in fact mostly cooler, rather than warmer (see Fig. 3a,b). Therefore, we conclude that the lateral advective heat flux, although present, is not sufficient to increase ground temperatures on the downhill side of the domain. Another mechanism must be active which causes

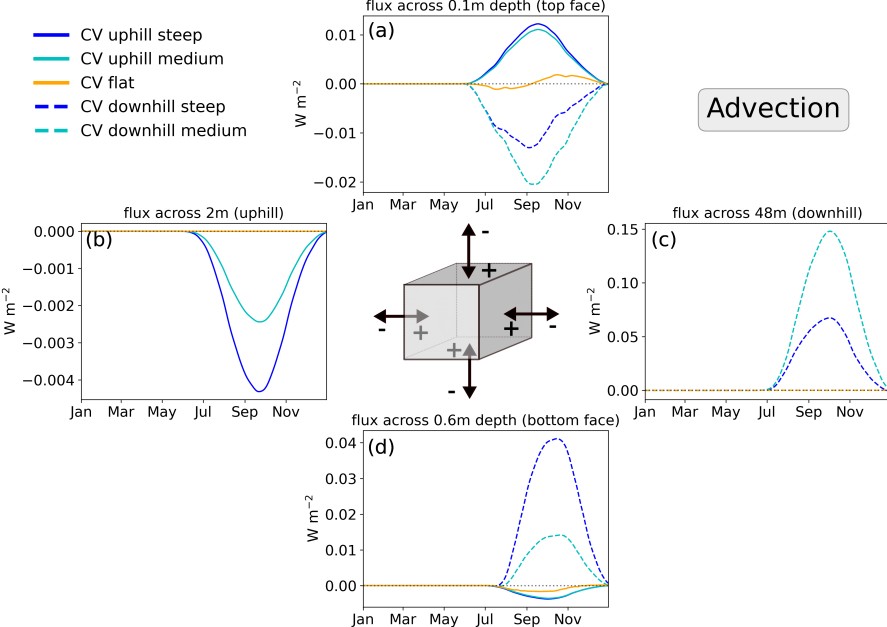

**Figure 8.** Daily values of advective heat flux on the faces of the control volume (CV; 90-day moving average) at the uphill (solid) and downhill (dashed) CV locations. Colors represent the steep (blue), medium (cyan) and flat (yellow) case, respectively. The sign convention adopted is positive values represent heat entering the CV and negative values leaving the CV. Due to the definition of the CV boundaries, lateral fluxes only occur on the right face for CV up and on the left side for CV down.

the downhill side to cool. This implies that the lateral flow of water, which carries heat, has a negligible effect on the warming towards downhill, based on the model configuration and hydroclimatic conditions used.

Finally, consider vertical advection across the near-surface face at -0.1 m depth (Fig. 8a), which is strongly influenced by the uphill vs. downhill side along the hillslope. While the flat case shows values varying around +/- 0.005 W m$^{-2}$ in summer, i.e. corresponding to negligible net heat flux, the sloped cases have consistently positive values on the uphill side (heat entering ground, solid lines) and consistently negative on the downhill side (heat leaving ground, dashed lines) during the same period. This gain and loss of heat on the top CV face ($z$=-0.1 m) can be explained by surface recharge (positive, i.e. heat gain) and evaporation (negative, i.e. heat loss). The positive heat flux on the uphill side is dominated by infiltration. As this is the driest part of the transect, it provides less moisture available for evaporative cooling. This energy flux directed towards the subsurface together with an overall lower heat capacity explains why the uphill part of the transect is warmer during summer (Fig. 4, upper panels). The negative flux on the downhill side is a result of higher liquid saturation providing more water for evaporation, which transports water and heat upwards out of the model (i.e. evaporative cooling). Evaporative flux as well as net infiltration (P-ET) directly at the surface ($z$=0 m) is given in Fig. S9 in the supplementary material. In deeper layers of the active layer (i.e. the bottom face), positive advective heat flux transports energy towards the surface (Fig. 8d, dashed lines),

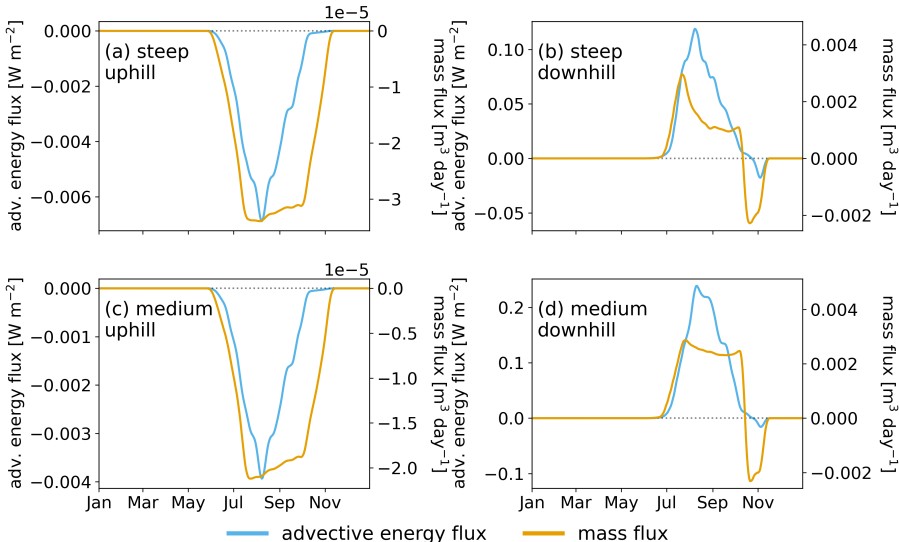

**Figure 9.** Daily lateral advective energy flux (blue) and mass flux (orange) through the vertical faces of the uphill **(a,c)** and downhill **(b,d)** CV. Daily values are averaged over a 7-day window. Note that the fluxes have different units. The sign convention adopted is positive values represent heat entering the CV and negative values leaving the CV.

which can explain the positive values (slope is warmer than the flat case) in Fig. 3d,f. In the uphill CV (Fig. 8d, solid lines), energy keeps getting transported down into deeper layers, contributing to warmer temperatures in the lower active layer.

### 3.5 Combined mass and energy fluxes

To further understand how much energy is carried by laterally seeping water, we compare the lateral advected energy flux on
the left or right faces of the CVs (corresponding to Fig. 8b and c) alongside the lateral water mass flux on the same faces, and compare the timing of peaks (Fig. 9; a complete presentation of the mass fluxes across all faces is provided in Fig. S10 in the supplementary material). Note that units between advective heat flux and mass flux are different and that the following interpretation focuses on the shape of the curves, rather than absolute values.

As can be seen (Fig. 9), advective heat flux (blue) peaks before September in both uphill and downhill CVs in both slopes
and declines shortly after. Mass flux (yellow) also has its first peak before September, but with prolonged duration of flow and declines more gradually. For the uphill CV, it can be seen that advective heat flux is close to zero already by October, while mass flux reaches zero only by mid November. The downhill CVs exhibit a second, less distinct mass flux peak just before and during freeze-up in the end of October, which is however not associated with a peak in advective heat flux.

The findings from both CVs indicate that heat is being carried with water flow during the warm season, corresponding to
410 mid-thaw period, but little advective heat is being transported by the end of the thaw season. This is caused by the permafrost acting as a significant heat sink and reservoir for cooling of the soil column above. Infiltrating water from the surface gets

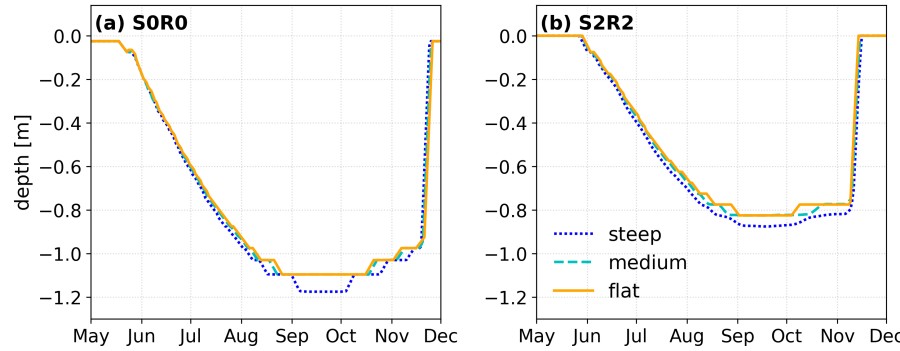

**Figure 10.** Representation of thaw depth compared between the steep (blue), medium (cyan) and flat case (yellow) as daily, spatially averaged thaw depth temporally averaged over a 5-day window from May to December in the last year of the simulation. Note that thaw depth is defined as cells within the model domain that exceed $0\,^{\circ}$C. **a** shows the results for the S0R0 (dry) scenario, while **b** shows daily thaw depths for the S2R2 (wet) scenario.

cooled down rapidly causing it to attain equilibrium with its surroundings. Then, although water seepage and flow occurs, it does not contribute much to advective heat transport, as the flowing water is at the same temperature as its surroundings. Note also that during freeze-up (November) in the downhill CVs, there are negative values for mass flux (Fig. 9b,d), indicating moisture is leaving the CV in the uphill direction, which we attribute to two-sided freezing and lateral cryosuction. While the active layer starts freezing from above, it also freezes from below, causing high water pressure in the remaining space occupied by liquid water. Due to the temperature distribution in the slope and valley bottom, the only direction the water can be squeezed out towards is uphill. Even though this effect might be overemphasized in a 2D domain, it is a physical based effect unique to permafrost landscapes. Additionally, unfrozen water in the downhill side of the domain can migrate towards the freezing front approaching from the uphill side (lateral cryosuction).

## 3.6 Impact of changes in precipitation

Due to the overall dry climate in Adventdalen, we conducted a sensitivity test to elaborate how the the model results change in a drier or wetter climate. Two additional wetness scenarios are considered for each hillslope; an even drier scenario (S0R0) and a scenario with increased wetness (S2R2). Snow (S) and rain (R) precipitation rates are set to 0 for S0R0, resulting in a completely dry climate, and the rates are multiplied by two in the S2R2 scenario, resulting in a climate that is twice as wet as the current climate. We compare the scenarios with regard to temperature differences, active layer thickness, and timing of freeze-up. Firstly, we find that both slopes and the flat case are notably warmer in the no-precipitation scenario (S0R0) and colder in the doubled precipitation (S2R2) scenario (Table S4 in the supplementary material). Relative temperature differences between the slopes and the flat case are generally in a similar range as in the original precipitation scenario. The steep slope in S2R2 is up to $0.7\,^{\circ}$C warmer than the flat case in summer and up to $-0.6\,^{\circ}$C colder in winter. In S0R0, the steep slope is up to $0.3\,^{\circ}$C warmer than the flat case in summer and up to $-0.2\,^{\circ}$C colder in winter.

Active layer thickness support these findings (Fig. 10). Maximum active layer thickness is deepest in the scenario with no precipitation (steep: 1.18 m, medium and flat: 1.1 m), while it is shallowest in the doubled precipitation scenario (steep: 0.88 m, medium and flat: 0.825 m). Note that the difference in absolute maximum active layer thickness between the medium and flat slope is very small averaged throughout the transect. Due to the temperature difference, however, the medium case experiences an earlier thaw and delayed freeze-up in the sensitivity scenarios as well as in the original scenario.

The timing of thaw and freeze-up is different throughout the inclinations in each scenario. In the original scenario, all cases start thawing by May 24 and are fully frozen again on November 23. In the scenario with no precipitation (S0R0), thaw has started in May 18 for the medium and flat case (the steep case on May 24), and freeze-up is complete on November 21. In S2R2 thaw begins on May 27 in the medium and flat case and May 29 in the steep case, while the last day with unfrozen subsurface-cells is November 11, almost two weeks earlier than in the other two scenarios.

Overall, the scenarios show that a higher amount of recharge added through precipitation on the surface will decrease the ground temperatures in the slopes as well as in the flat case. Note that multiplying snow by a factor of 2 still did not result in a snow cover significant enough to have an insulating effect on the subsurface.

We attribute the temperature difference of the original scenario to the effect of changes in heat capacity and increased/decreased moisture availability for evaporative cooling in the wetter and drier scenario, respectively (not shown). These results are consistent with previously observed cooling effect of precipitation on the active layer. Wen et al. (2014) and Wu and Zhang (2008) both documented a cooling of the active layer in response to rainfall on the Tibetan Plateau. In contrast, e.g., Douglas et al. (2020) and Mekonnen et al. (2021) found a warming effect of summer precipitation on active layer temperatures. However, those studies did not account for the influences of topography.

These findings imply that potential future changes in air temperatures and precipitation towards a warmer and wetter climate could have opposing effects on subsurface temperatures. While higher summer temperatures have a high potential to increase active layer thickness in a catchment, higher precipitation amounts could counteract these processes and act as a heat sink.

Therefore, the interaction of warmer temperatures and increased precipitation rates under changing climates warrants investigation. Moreover, a transient development of a combined temperature and precipitation scenario is likely to yield a different result than our step-wise increase of precipitation alone. Potentially, a deeper active layer might lead to a greater volume of unfrozen soil and water, which is available for energy transport (Walvoord and Kurylyk, 2016). This could then lead to even higher non-linearly increasing advective heat fluxes that could eventually contribute to the energy budget downhill.

## 3.7 Outlook

Advancements in 2D permafrost modeling have previously shown that lateral flow of water and associated advection of heat in sub-Arctic, discontinuous permafrost landscapes can significantly change the temperature regime of the subsurface as well as the timing of thaw and freeze-up (Sjöberg et al., 2016). Shojae Ghias et al. (2019) and McKenzie and Voss (2013) also showed in several model setups that a combined conduction-advection scenario causes an increased permafrost thaw as opposed to a conduction-only scenario, highlighting the importance of lateral heat advection. In a polygonal tundra, continuous permafrost landscape setup, model results by Abolt et al. (2020) show that temperature differences within a single polygon are caused

by moisture redistribution. While the rims were drier and warmer, the centers showed colder temperatures. This is attributed to heat capacity and evaporative cooling, which is low in dry areas and high in wet areas. Accordingly, lateral energy fluxes are governed by lateral conduction and temperature gradients. Even though this is on a much smaller scale than our hillslope simulations, it shows similar governing effects of temperature distribution as in the present study and highlights the importance of lateral processes not only in the form of heat advection. Evaporative cooling has previously been identified as one of the major non-conductive heat fluxes causing a subsurface cooling in permafrost landscapes (Kane et al., 2001; Wu and Zhang, 2008; Wen et al., 2014; Li et al., 2019; Luo et al., 2020).

The observed temperature differences between uphill and downhill of up to about 0.80 °C for steep (22°) and 0.56 °C for medium (11°) slopes in the present study is obtained for a model domain with a lateral distance of 50 m. We generalize these results by calculating lateral ($x$-direction) and vertical ($z$-direction) cooling rates based on the slope inclinations. For the steep slope case, this results in a lateral cooling rate of 0.016 °C/m and a vertical cooling rate of 0.04 °C/m. For the medium slope, the lateral cooling rate amounts to 0.01 °C/m. The vertical cooling rate is higher (0.056 °C/m) than in the steep slope case. These rates are representative for slopes in the Adventdalen area in Svalbard under current climatic conditions.

Projecting these results to larger scales, hillslope processes might cause significant differences in permafrost distributions throughout a catchment. As shown in Fig. 1, slope inclinations described in this study are present in almost all regions throughout the Arctic and therefore should be accounted for in larger-scale permafrost models. Besides Svalbard, other regions such as Greenland, Yukon, and the Russian Far Eastern Federal District show a considerable share of slopes within the steepness-range simulated in this study. Since our slopes were limited to 50 m in horizontal distance, it can be expected that longer slopes enhance desaturation uphill and aggregate more water towards the downhill side, eventually leading to fully saturated conditions and surface water formation at the slope base. At the same time, lateral advective heat fluxes have shown to increase non-linearly with increasing precipitation, which might also be observable in larger scale hillslope systems due to higher water availability. Considering a full, 3D representation of a hillslope, it is likely that the micro topography within the slope causes further concentration of moisture, eventually leading to water tracks, which have shown to act as conduits for groundwater even if the adjacent hillslope is already frozen (Evans et al., 2020). These features might substantially change the observed effects in this homogeneous 2D representation of a hillslope without micro topography.

Furthermore, applying this model in a wetter environment or considering potential climate change scenarios towards a wetter climate, new effects of water redistribution might become visible. Ponding water on the downslope side of the domain or in the valley bottom can start forming a talik when energy requirements for the phase change from water to ice (latent heat) become too high. At the same time, higher thermal conductivity leads to greater heat loss towards the subsurface. These competing effects have been studied by Atchley et al. (2016) in a 1D column model, and found that these processes potentially cancel each other out. Clayton et al. (2021) also found that both these processes can be active at the same time in different depths. Furthermore, considering the shallow snow cover in the present study, a potential greater snow cover can lead to insulation effects, which can *(i)* further increase the effect of uphill warming by insulating the overall warmer soil from cold air temperatures, *(ii)* provide more water to the subsurface during snow melt and increase evaporative cooling also in the uphill

part of the slope, and/or *(iii)* insulate a potential talik in the downhill part of the domain if enough liquid water is available in summer.

## 4   Conclusions

This study shows that there are differences in the thermal-hydraulic state of the subsurface between the uphill and the downhill side of a 50 m long hillslope transect, with the uphill area being warmer and the downhill area being colder when compared to
each other. Vertical advective heat fluxes (infiltration and evaporative cooling) and both heat capacity and thermal conductivity play a major role in this comparison causing a great share of the differences between the flat control case and the sloped cases. The warming effect is strong enough to increase end-of-season active layer depth by 5.5 cm between the flat and the steep case. Based on the objectives and investigation questions outlined in this study, the conclusions are as follows.

*(i)* Hillslope inclination causes differences in ground temperature uphill and downhill. We found that upill sides are generally
warmer than downhill sides. This uphill warming effect is up to about $0.80\,°C$ for steep ($22°$) and $0.56\,°C$ for medium ($11°$) inclinations across a lateral distance of 50 m representative for valleys in Adventdalen, Svalbard.

*(ii)* The steep slope causes ground warming on the uphill section strong enough to increase maximum active layer depth by 5.5 cm (1.03 m) as compared to the flat case (0.975 m). The medium slope only incurs sufficient uphill warming to increase maximum thaw depth by 1.1 cm (maximum active layer depth is 0.986 m) compared to the flat case. However, the total volume
of unfrozen soil during the warm season increased by 1.7% in the steep slope case, and 0.6% in the medium slope case.

*(iii)* The uphill warming and slight downhill cooling phenomena observed here are determined to be caused by three main processes:

1. Higher moisture content downhill than uphill due to gravitational flow and water accumulation, which increases effective thermal conductivity and associated heat loss to the atmosphere in the downhill section in winter; also drying in the uphill
section slightly reduces less heat loss in winter.

2. Reduced moisture content in the uphill section decreases effective heat capacity, which leads to faster warming, while increased moisture content in the downhill section increases heat capacity and slows down summer warming.

3. In summer, increased moisture content downhill increases evaporation and leads to greater evaporative cooling; in the uphill section, evaporative cooling is limited by the dry conditions, leading to relative heat gain through infiltration
compared to the downhill side.

We find that the temperature differences do not linearly increase with a linear increase in slope inclinations ($11°$ to ($22°$)) and do not double between the two sloped cases. While active layer thickness increases by more than 5 cm between the steep and the flat case, the medium slope only experiences a 1 cm deeper active layer. This finding, although based on numerical physically-based modeling, should be observable in field conditions for this type of environment and hydroclimatic conditions.
It highlights the relevance of considering lateral flow of water in the subsurface combined with heat flux for modeling arctic

catchments with permafrost. It also has implications for interpretation of thermal measurements and time series logging in hillsopes.

*Code and data availability.* The Advanced Terrestrial Simulator(ATS) (Coon et al., 2019) is open source under the BSD 3-clause license and is publicly available at https://github.com/amanzi/ats (last access: July 2020). Simulations were conducted using version 0.88. Forcing
datasets and input files are available at https://github.com/a-hamm/ats_hillslope2021.git. Weather data to create the forcing dataset was downloaded from the UNIS website (https://www.unis.no/resources/weather-stations/) (last access March 2020) and from the Norwegian Meteorological Institute (https://www.eklima.met.no) (last access March 2020).

*Author contributions.* Both authors conceived the initial conceptualization. Model simulations were performed by AH with guidance from AF. AH wrote the manuscript with contribution from AF.

*Competing interests.* The authors declare that they have no conflict of interest.

*Acknowledgements.* This work is funded by Formas (project 2017-00736) with support from the Bolin Centre for Climate Research. We highly appreciate the valuable comments from two anonymous reviewers, which greatly helped to improve the quality of the work. The authors also thank Ahmad Jan and Ethan Coon for technical support with ATS.

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
