# Peer review of "Impact of lateral groundwater flow on hydrothermal conditions of the active layer in a high arctic hillslope setting"

_The Cryosphere, 2021_

## Author Response (AR1)

**Response letter to Referee #1**

Our responses are written in blue font.

(Examples of) line numbers in the **revised manuscript** that contain changes according to the comment are given in red font.

**General Comments**

1.      As stated on L135-136, the "downhill end of the transect represents the valley bottom and allows for water accumulation and potential ponding on the surface". According to Figure 2, this downhill boundary is a no-flow and no-heat flux boundary but there is little justification for this no-flow condition. It looks like in Figure 1c that these hills ultimately flow into a river so are instead, flow boundaries. It may make more sense to represent these as flux or constant pressure boundaries. As no-flow boundaries, I worry that they are artificially blocking heat transport and accumulating water, which is why the rightmost column in Figure 4 does not make sense with the rest of the modeled cross-section. If these no-flow boundaries are not affecting the model outputs, it would be helpful to see a comparison of model outputs with and without the no-flow downstream boundary in the supplements. If this downslope no-flow boundary does change results, please revisit much of the results text, including your third conclusion.

Response: Thank you for this important comment. The use of no-flow and no-heat flux boundaries on the vertical sides of the domain is intentional. The conceptualization is that the no-flow boundary condition represents a watershed boundary on the uphill side and a symmetry boundary condition on the downhill side, the latter corresponding to flow accumulation from both sides of a symmetric V-shaped hillslope valley transect. We realize this was not clearly stated in the manuscript and have made efforts to clarify this in the revised model description section.

We are not attempting to simulate a valley that ends in a perennial stream or river, where a constant head boundary could be suitable. Instead, the surface mesh of the model allows for ponding and ice/snow accumulation to occur. Surface water ponding can be combined with a spill-over threshold condition to limit excessive ponding, thereby corresponding to a maximum depth of a surface water body. This implementation thereby avoids the need for a constant head boundary, and allows for surface water to form as a result of the flow and energy balances and the hydro-meteorological forcing of the system. The maximum depth of the surface water body is then the height of the spill-over condition. However, in our simulations, no surface ponding occurs. This is due to the relatively dry conditions of the site and because evapotranspiration allows for much water to be removed from the system prior to saturating the active layer.

However, we acknowledge that a V-shaped geometry is not so representative for typical valley transects on Svalbard. We have therefore redesigned the model domain to resemble a simplified U-shaped geometric conceptualization, by extending the mesh to account for a flat valley bottom subsequent to the slope (see Fig. 2 in the main text and Fig.1 in this letter); the flat part is an addition of 16 meters, represented by 8 mesh elements/columns). This means that heat and water can now move out of the lowermost slope-column into the flat part of the domain, corresponding to the valley bottom.  The same no-flow boundary conditions are applied to the rightmost vertical edge in order to produce a symmetry boundary for the downhill side, and as before,

ponding and ice/snow accumulation is allowed to occur on the surface mesh of the domain. For the analysis and presentation of results, we still only consider the first 50m of the domain (the sloped part of the domain).

We have evaluated potential boundary effects of proximity to the right edge of the domain (the downslope side) by examining yet another domain configuration, consisting of a flat extension of only 4 meters (2 columns). That case yielded similar results as the 16-meter extension used in the main revised simulations. Thus we conclude that a 16-meter extension for representing the valley bottom is more than sufficient to be a safe distance from the right symmetry boundary and to avoid any boundary/edge effects.

This revised model domain configuration has changed some of our results, specifically they show a reduced moisture accumulation on the downhill side compared to the previous model, and subsequently a dampened cooling effect in the downhill section. However, the uphill warming effect is largely unchanged. We have adjusted the text and figures in the manuscript accordingly.

[Figure]

*Figure 1: Conceptual representation of the surface and subsurface modeling domain. Grey shaded areas on either side of the transect indicate the uphill and downhill observation locations, red indicates the sides of the model, blue boxes represent the control volumes (CV) and a blueline at the bottom indicates the bottom boundary. Thermal-hydraulic boundary conditions (BC) on the surface, sides and bottom are listed on the right.*

E.g.: L144-147 and L160-166

2.      The presented models are referred to as idealized but are based on field data from Adventdalen, Svalbard which makes me wonder why a calibration was not performed? I think that uncalibrated models can be useful thought experiments, and I understand that calibrating and validating a model can be taxing. However, I question the validity of using conclusions from a model that is not calibrated to existing field measurements, especially using a model that is in the middle

ground between an uncalibrated generalized model and a model that is calibrated. At a minimum, the authors need to suggest how the model results may compare to field observations of similar sites.

Response: Indeed, our model is in the middle ground between an idealized and a site-specific/calibrated model. Since we do not have subsurface data from any of the slopes in the area, it is not possible to conduct a proper calibration. However, as mentioned in the text, a similar site in the Adventdalen valley bottom (the UNISCALM site) has been investigated previously by Schuh et al. (2017). We refer to this as a 'similar site' and expect similar results for our reference case (with no slope). In the study by Schuh et al. (2017), model results showed good agreement with subsurface observations (especially temperature and thaw depth) over a time period from 2000 to 2014, even though model parameters were derived from literature. Furthermore, their simulated active layer depths of around 100cm is consistent with the measured active layer depths in that area (see technical comment no. 5). In our simulations we also obtain similar active layer depths. Since the time duration of the subsurface observation dataset of the UNISCALM site does not overlap with the available hydro-meteorological dataset, a calibration for the flat model case is not realistic. We have added a clarification in the paragraph that describes the material properties, which indicates that these properties have been found to accurately represent the subsurface at the UNISCALM site.

E.g.: L155-159

3.     It is unclear how relevant these findings are to permafrost landscapes throughout the Arctic. How often are there hillslopes of a constant slope without valleys and lateral (cross hillslope) water flow? Even more basic, what percent of the Arctic is sloped terrain? Any additional information that could be provided to aid in the upscaling of these results outside of Svalbard would be beneficial.

Response: Thank you for this great idea. To address the question how representative the slopes are throughout the Arctic, we studied slopes in different regions (countries with administrative areas in the Arctic) around the Arctic based on a digital elevation model (ArcticDEM, 10m resolution). To perform the necessary calculations, we developed an algorithm which classifies the slopes according to four categories of different inclination with GIS software. We only consider areas of continuous permafrost in this evaluation. A figure with the results that has also been incorporated in Fig. 1 in the main text can be found below (Fig. 2 in this letter).

We added information on the methodology as well as the results for the individual pie charts into the supplementary information of the revised study. We found that, as expected, most of the landscape is comparably flat (0-5°). However, slopes in both categories defined in our paper, 5-15° and 15-25°, are represented in every region and range between 12-30% and 2-14%, respectively. Regions, which stand out in terms of significant proportion of land area with moderate to steep slopes include Yukon, Greenland and the Far East Federal District in Russia, together with Svalbard. Thus we feel it is important to study permafrost and active layer dynamics in hillslopes, certainly for Svalbard, but also for the Arctic in general.
Furthermore, we did not investigate complex topographies like cross hillslope water flow as the idea is to simulate a hillslope along a preferential flow pathway.

[Figure]

*Figure 2: Classification of slopes around the Arctic according to the classification in this manuscript. Grey areas show land masses, blue areas indicate continuous permafrost areas and each pie chart represents a different administrative region in the Arctic. Slopes have only been calculated for areas of continuous permafrost and within the extent of the ArcticDEM.*

E.g.: L132-137 and L454-457

**4.** The assumptions of the study, especially the modeling assumptions, should be specifically stated in a separate section. For instance, this simulation doesn't include an organic layer, but organic layers exist in many permafrost landscapes and have very different thermal properties from mineral soil (this goes back to if these results can be upscaled or not). Is it reasonable to model hillslopes in only two-dimension? I think it can be but the reason for doing so needs to be stated and supported with other peer-reviewed papers.

Response: Thank you for this comment. We agree that the justification about the model domain setup should be clarified; we have made efforts to improve this in the revised manuscript, specifically we added an additional study about 2D slope simulations to support our study (Jan and Painter 2020 and Jafarov et al. 2018).

Even though it is correct that an organic layer is common in permafrost landscapes, we decided to exclude this from our simulations for mainly two reasons. First, the material along slopes (at least in Svalbard) is often much less organic than in the valley bottoms and sometimes completely absent (depending on the steepness), and we wish to keep identical subsurface textures between the different slope inclination cases to enable consistent comparisons. Second, we briefly looked at the effects of organic layers on our preliminary investigations and saw that it made substantially more difficult to understand and untangle different processes; including subsurface heterogeneity would be beyond the scope of this analysis but we are considering pursuing this line of investigation in future efforts.

E.g: L156-157 and L151-152

5.    The main text needs to be revised for clarity. The figures are attractive and easy to see, which is appreciated, but many of the figures need additional annotations or subfigures to help with comprehension. It is also unusual to have the results and discussion sections combined. I would highly recommend separating these sections so you can have a more thorough discussion section where you interpret your results and compare them to other peer-reviewed studies. As is, this combined section is quite long and hard to digest. I have pointed out some specific examples below where the text and figures need to be revised for clarity.

Response: Thank you for this feedback. We have opted for combining text on results and analysis/discussion because we find this to be the most efficient and clear way to present and understand these findings. This study uses a physically-based numerical model to both analyze and explain system behavior, and due to the coupled nature of many of the physical processes and quantities we feel the most convenient and clear way to understand the findings is by a combined presentation of results with analysis and discussion.
We have made efforts to improve the text and presentation by incorporating the specific and technical comments below and restructuring the text where needed for better readability. We have also revised some figures and several of the figure captions, to help improve the clarity, and included additional information, which we hope helps to interpret them more easily.

See line numbers in the following comments

**Specific Comments**

1. L6, How representative are these hillslopes of Arctic landscapes as a whole?

Response: We have analyzed hillslope inclinations of Arctic landscapes, please see our response to the general comment no. 3 above.

L132-137

2. L15, Since this study only considers one slope versus a 'hilly' landscape, I would hesitate to draw this conclusion about hilly terrain.

Response: Replaced 'hilly' with 'sloped'

3. L29, Rather, permafrost degradation has been found to increase groundwater discharge into surface waters, not decrease the seasonal variability.

Response: Permafrost degradation has been shown to specifically reduce seasonal variability in groundwater discharge, please see:

Frampton, A., Painter, S., Lyon, S.W., Destouni, G., 2011. Non-isothermal, three-phase simulations of near-surface flows in a model permafrost system under seasonal variability and climate change. Journal of Hydrology 403, 352–359. https://doi.org/10.1016/j.jhydrol.2011.04.010

Frampton, A., Painter, S.L., Destouni, G., 2013. Permafrost degradation and subsurface-flow changes caused by surface warming trends. Hydrogeology Journal 21, 271–280. https://doi.org/10.1007/s10040-012-0938-z

To clarify this statement the citations have been updated in the revised manuscript.

No changes.

4. L41, How much topography is 'more topography'? Is there are slope cutoff? Be specific.

Response: Removed 'and landscapes with more topography'

L41-42

5. Section 2.1, What are typical active layer depths at the study site?

Response: Added a sentence: Active layer depth in the area ranges from 90 to 110cm (measured between 2000 and 2018; Strand et al., 2020)

L107-108

6. L107, How far is this from the weather station in km?

Response: Added information in brackets: *Precipitation data was retrieved from the long-term weather station at Longyearbyen airport (9 km west of the Adventdalen weather station; 78.24◦N 15.51◦E)*

L112-113

7. L118, What is the hillslope length?

Response: Added information in brackets: *To inform the model, the same forcing dataset is used for the entire model domain (50m transect length) without accounting for temperature lapse rates between the lower and upper part of the transect.*

L120-122

8. Figure 2, It would be helpful to show node locations.

Response: We agree that the node location is an interesting additional information. Unfortunately, the mesh resolution is too high to meaningfully include it into the original figure (Fig. 2 in the main text) without making it excessively cluttered. However, we will include a figure showing the three different meshes in the supplementary material, and also provide the mesh files (.exo format) in the data repository.

L150-151 and 507-511

9. L136, What is the depth of the mineral soil?

Response: The depth of mineral soil is 20m, which is the domain depth (homogenous throughout the domain). We added a clarification of this, as follows: *All cases assume a homogeneous material throughout the model domain consistent with mineral soils typically encountered in the area.*

L155-156

10. Table 1, Thermal conductivity values are low for a mineral soil, I would expect closer to 2.7 W/mK.

Response: Thank you for catching this. This was indeed a mistake, as we intended to use the same material properties as in Schuh et al. (2017). We have re-run the simulations with liquid saturated thermal conductivity = 1.7 W/mK , and dry thermal conductivity = 0.27 W/mK (following Woo 2012 (Table 2.1) as described in Schuh et al. 2017) and updated the parameters in Table 1. This correction has changed our results only slightly; we now obtain deeper active layer depths which in fact are even closer to typical measured active layer depths of the general valley bottom region in Adventdalen, as presented by Strand et al. and Schuh et al. (around 1m).

Table 1

11. L140, What makes the no flow lateral (right and left) boundaries? Are they a watershed divide? Seems unlikely given the flat lateral topography.

Response: Thank you for your question. Indeed, the vertical boundary on the upper (left) side represents a water divide, while the vertical boundary on the lower (right) side represents a valley bottom and is a symmetry boundary for flow accumulation (and not necessarily ending in a surface water body), please see our response to general comment 1 above. Given the length of the slope-transect (50m), the definition of a water divide on the upslope boundary might not be applicable to all slopes but serves as a suitable compromise between conceptual representation of a hillslope and computational effort for analysis. We further discuss this issue in the outlook section (Section 3.7). To clarify this, we have added a sentence in the simulation configuration section

L160-166

12. Figure 3, I'm confused about what these plots are showing. It would help with clarity to first plot the temperature time series for the steep, medium, and flat simulations for both uphill and downhill and then plot these differenced values. Would also be helpful to annotate this figure, for example, you could write 'uphill warmer' above the x-axis in (a).

Response: We included additional information in the figure caption to help guide the interpretation. Unfortunately, it is not possible to include the time series in the main figure, but we agree that it is useful to see it as well. As a solution we added a figure (Figure S3) including the time series in all depths for up- and downhill locations in the supplementary information.

L215-216

13. L192, What about the large November peak at 0.75 m in Figure 3a?

Response:  The temperature differences in November are particularly high close to the permafrost table as this part of the active layer freezes faster and gets overall colder. This is likely due to higher thermal conductivity when air temperatures drop in winter and the presence of the permafrost, which is acting as an additional heat sink. We added a sentence in section 3.1 to describe this in more detail.

14. Figure 4, Label color bar with units. Which is the uphill side? Horizontal distance=0 m? Label this on the figure. Why is the far-right column in each subfigure so different than the other columns? It looks like a potential modeling error due to the no-flow boundary conditions that do not physically make sense. This difference was pointed out in section 3.2 but is concerning. Also, are the two dotted lines in the October subfigures showing the presence of a horizontal talik?

Response: We added labels for the color bars and added two labels on the x-axis indicating up- and downhill, where x=0 m corresponds to the uphill side and x=50 m to the downhill side. The last column receives water from uphill and therefore exhibits different behaviour; however we have now changed the geometry of the domain by including a 'valley bottom' and have addressed concerns of potential boundary effects, please see our response to general comment 1 above. A horizontal talik during freeze-up is indeed present and indicates two-sided freezing, and the December image of Figure 4 in the main text shows that this talik is not permanent; this is now clarified in the main text.

15. L279-280, This is likely due to the no-flow boundary on the downslope side, and is not realistic if there was a river or otherwise at this boundary.

Response: Please see our response to general comment 1 above, where we address the concerns of domain geometry and boundary conditions.

16. L284-286, What about the role of specific heat, where specific heat is higher for saturated soil than unsaturated soil? This may explain your results on L390-391.

Response: Thank you very much for this suggestion, we agree that it is a potentially important factor. We unintentionally overlooked discussing heat capacity. As pointed out in the comment, moisture distribution leads to differences in saturation between the up- and downhill sides of the domain, which in turn leads to increased heat capacity in wetter areas and decreased heat capacity in dryer areas. This is now added to Figure 6 in the main text. As a consequence of differences in heat capacity, dryer areas both warm up and cool down faster, because less heat is needed to cause a temperature change (on either side of the zero degree curtain). We also added an additional figure in the supplementary information (Fig. S5) that shows differences in heat capacity in a 2D plot analogous to Figure 4 in the main text. We adjusted the text in several places to incorporate this information.

17. L326, Vertical diffusion of what? Heat diffusion? Be explicit even though you go on to reference heat.

Response: We added 'heat' for all the different occasions of 'diffusion' in the text.

18. L430-440, Why are these processes more relevant for a high-Arctic hillslope setting if they are for sites with no topography?

Response: We realize that the formulation in this paragraph was not clear. We rewrote parts of the paragraph to clarify our intended meaning. The main message we wish to convey relates to the difference in processes between a discontinuous permafrost landscape versus a continuous permafrost landscape; while heat advection is found to alter the temperature regime in discontinuous permafrost landscapes (Sjöberg et al., 2016, Shojae Ghias et al. ,2019), advection of heat does not play a significant role in continuous permafrost landscapes with a similar topography and small hydraulic gradients (Kurylyk et al., 2016). The difference to our hillslope study is that the distribution of water actually impacts the temperature differences between two locations along the slope (uphill vs. downhill) rather than the advection of heat itself.

L434-446

**Technical Comments**

19. L2, What is 'its'? Permafrost?

Response: Since the sentence was a bit confusing, we changed it to: *Modeling the physical state of permafrost landscapes is a crucial addition to field observations in order to understand the feedback mechanisms between permafrost and the atmosphere within a warming climate.*

L2

20. L4, Delete 'want to'.

Response: Deleted.

L5

21. L6, Indicate that these are the 'steep' and 'medium' cases.

Response: Adjusted.

L7

22. L50-54, These sentences seem out of place and are too short to form a paragraph. Add more studies, incorporate them into the previous paragraph, or remove them.

Response: We have merged the paragraph to the end of the previous paragraph. This part is intended to highlight processes in the high Arctic, as compared to studies mentioned before that mainly focus on sub-Arctic systems.

L50-54

23. L55, What is the length of the hillslopes? I imagine hillslope length will alter results.

Response: Slope/transect length is mentioned later on in model design but added now as well to the end of the introduction part. Slope length is also an important factor to consider, which is addressed in the outlook section (section 3.7)

L120-122 and L457-459

24. L55, Add 'two-dimensional'.

Response: Added.

L62

25. L54-73, Condense into one paragraph, move study site details to study site section.

Response: Condensed paragraphs into one paragraph and moved information about the location of Svalbard and respective active layer deepening to the study site section.

L107-110

26. L68-69, Delete, repetitive.

Response: Deleted.

L67-68

27. L71, To 'what' extent.

Response: Adjusted.

L70

28. L71, I think replacing 'inclination' with 'slope' would make this easier to understand as it uses the more common term.

Response: Thank you for your suggestion. However, we played around with different word combinations and we think that hillslope inclination should be clear enough for readers to understand that the different inclinations (11 and 22 degrees) refer to the 'slope' of the hillslope.

No changes

29. L72, Typo, 'to'.

Response: Corrected.

L70

30. L89 and L93, Citation typo.

Response: Corrected.

L91 and L95

31. L112, This is the mean snow and rain for 2013-2019, correct? It's unclear as written.

Response: We clarified the sentence: *The resulting average sum of rain (160mm) and snow (170mm water equivalent, total precipitation=330mm) for the period 2013-2019 was then redistributed to equal daily amounts during the rain- and snow period, respectively.*

L117-119

32. L118-119, Change to 'slope'.

Response: Changed.

L124

33. L149, What does 'field values' mean here?

Response:We rephrased the sentence to: *The model output is given as cell values in selected cells of the model domain.*

L178

34. L162, Typo 'initialization'.

Response: Corrected.

L192

35. L186, Delete 'significant' are these trends statically significant?

Response: Removed.

L217

36. L187-188, Highlight these times (thaw and freeze up) in Figure 3 with shading or otherwise.

Response: Added shading to Figure 3.

Figure 3

37. Table 2 and 3, Can move to supplementary material.

Response: Moved to supplement and changed text to refer to tables.

L231-232

38. L199-203, Redundant, can be removed.

Response: Removed.

removed, see comment 37 above

39. L205-206, I don't think you can draw this conclusion from the presented data, save this for the discussion.

Response: Agreed. The text has been reworked and moved in the revised manuscript.

Text has been removed

40. L207, What do you mean by 'inversion of temperature differences'? Please reword.

Response: Sentence has been removed with paragraph (in response to comment 39 above).

Text has been removed

41. L207, Again, I don't think this necessarily indicates this conclusion, remove.

Response: Agreed, removed

Text has been removed

42. Figure 5, Delete panel (b), panel (a) is clearer and shows a very similar result.

Response: Thank you for your suggestion. After revisiting the figure and the corresponding text, we agree that the Figure 5b does not add anything new to the results. We therefore removed that panel and modified/simplified and partly removed the corresponding text accordingly.

43. L279, I'm not sure what lateral gravitational water flow means exactly since water flows vertically due to gravitational attraction.

Response: Removed 'lateral'.

L284

44. Throughout, Refer to as 'heat' diffusion.

Response: Added 'heat' to the occurrences of diffusion.

Added 'heat' to all occurrences of 'heat diffusion'

45. L337, This is an important point to make.

Response: Agreed.

L352-355

46. Section 3.7, Add a more descriptive header.

Response: Changed to 'Outlook'.

Section 3.7

47. L435-436, Also see McKenzie and Voss, 2013.

Response: Added McKenzie and Voss, 2013 for literature comparison.

L436

**References cited in this letter:**

Frampton, A., Painter, S., Lyon, S.W., Destouni, G., 2011. Non-isothermal, three-phase simulations of near-surface flows in a model permafrost system under seasonal variability and climate change. Journal of Hydrology 403, 352–359. https://doi.org/10.1016/j.jhydrol.2011.04.010

Frampton, A., Painter, S.L., Destouni, G., 2013. Permafrost degradation and subsurface-flow changes caused by surface warming trends. Hydrogeology Journal 21, 271–280. https://doi.org/10.1007/s10040-012-0938-z

Jan, A. and Painter, S. L.: Permafrost thermal conditions are sensitive to shifts in snow timing, Environmental Research Letters, 15, 084 026, https://doi.org/10.1088/1748-9326/ab8ec4, https://iopscience.iop.org/article/10.1088/1748-9326/ab8ec4, 2020.

Jafarov, E. E., Coon, E. T., Harp, D. R., Wilson, C. J., Painter, S. L., Atchley, A. L., and Romanovsky, V. E.: Modeling the role of preferential snow accumulation in through talik development and hillslope groundwater flow in a transitional permafrost landscape, Environmental Research Letters, 13, 105 006, https://doi.org/10.1088/1748-9326/aadd30, http://stacks.iop.org/1748-9326/13/i=10/a=105006?key=crossref.ea8d38a9a41cbb120144acdd5d1d4d37, 2018.

Kurylyk, B. L., Hayashi, M., Quinton, W. L., McKenzie, J. M., and Voss, C. I.: Influence of vertical and lateral heat transfer on permafrost thaw, peatland landscape transition, and groundwater flow: Permafrost thaw, landscape change and groundwater flow, Water Resources Research, 52, 1286–1305, https://doi.org/10.1002/2015WR018057, http://doi.wiley.com/10.1002/2015WR018057, 2016

Schuh, C., Frampton, A., and Christiansen, H. H.: Soil moisture redistribution and its effect on inter-annual active layer temperature and thickness variations in a dry loess terrace in Adventdalen, Svalbard, The Cryosphere, 11, 635–651, https://doi.org/10.5194/tc-11-635-2017, https://www.the-cryosphere.net/11/635/2017/, 2017.

Shojae Ghias, M., Therrien, R., Molson, J., and Lemieux, J.-M.: Numerical simulations of shallow groundwater flow and heat transport in continuous permafrost setting under impact of climate warming, Canadian Geotechnical Journal, 56, 436–448, https://doi.org/10.1139/cgj-2017-0182, http://www.nrcresearchpress.com/doi/10.1139/cgj-2017-0182, 2019.

Sjöberg, Y., Coon, E., K. Sannel, A. B., Pannetier, R., Harp, D., Frampton, A., Painter, S. L., and Lyon, S. W.: Thermal effects of groundwater flow through subarctic fens: A case study based on field observations and numerical modeling, Water Resources Research, 52, 1591–1606, https://doi.org/10.1002/2015WR017571, http://doi.wiley.com/10.1002/2015WR017571, 2016.

Strand, S. M., Christiansen, H. H., Johansson, M., Åkerman, J., and Humlum, O.: Active layer thickening and controls on interannual variability in the Nordic Arctic compared to the circum-Arctic, Permafrost and Periglacial Processes, p. ppp.2088, https://doi.org/10.1002/ppp.2088, https://onlinelibrary.wiley.com/doi/10.1002/ppp.2088, 2020.

Woo, M.: Permafrost Hydrology, Springer Berlin Heidelberg, Berlin, Heidelberg, 2012.

**Response letter to Referee #2**

Our responses are written in blue

(Examples of) line numbers in the **revised manuscript** that contain changes according to the comment are given in red font.

**General comments**

1. Based on what I read, the main message is because of variable saturation, hillslopes experience different thermal regimes with warmer upslope areas and cooler down slope areas. The paper seems to focus on how increased moisture in the down slope area causes increased evaporative cooling. However, the overall thermal state of the domain is slightly warmer. I'm left wondering why are sloped simulations overall warmer? Only at the end of the manuscript when the authors do an additional sensitivity study by adding more precipitation do we see a general cooling effect of the entire domain. This then suggests that it is a water balance mechanism caused by slope, i.e the system tips the evapotranspiration into an energy limited system rather than water limited and as more water is added. And that area of the domain with more evaporative cooling outweighs the domain that is water limited. However, the water balance verses degree slope, which appears to control the overall thermal state of the entire domain is not discussed.

Response: Thank you for the valuable comment. We realize that the downhill cooling effect received too much attention and that the overall warming of the slope as compared to the flat case has fallen short. We explain the uphill warming effect by increased infiltration (heat advection), less evaporative cooling and overall lower heat capacity as compared to the flat case. In the text, we briefly mention that vertical heat advection plays a major role in the differences between the up- and downhill section. While the uphill experiences less evaporation and more infiltration, the downhill experiences notably more evaporative cooling . We added a figure with a simple water balance (difference between precipitation and evaporation; P-ET) below (Fig. 1 in this letter and Fig. S6c and d in the supplementary material). It shows that overall evaporation is higher in the downhill side and lower in the uphill side (Fig. 1a,b in this letter), which in turn leads to positive values in the summer water balance (P-ET) in the uphill side (Fig. 1c in this letter), i.e. more rain than evaporation, and negative values in the summer water balance on the downhill side (Fig. 1d in this letter ), i.e. more evaporation than rain.

E.g. L374-378 and L490-499

[Figure]

*Figure 1: Evaporative flux (**a** and **b**) and net infiltration (precipitation-evaporation; **c, d**) at the surface on the uphill (solid lines; **a** and **c**) location and the downhill (dashed lines; **b** and **d**) location. Daily values are averaged over a 7-day window. Blue, cyan and yellow represent the steep, medium and flat case, respectively.*

2. Similarly – and at a much smaller scale, Abolt et al., (2020) found (using the same ATS model) that rims in polygonal ground warm more in the summer due to drier conditions and associated weakened evaporative cooling, which then provides energy laterally to the cooler saturated troughs in the summer (see section 5.2 of Abolt et al., 2020). However, what determines if a saturated area is cooler or not also depends on the mass of water present.  If enough water is present, especially on the surface, then a Talik will begin to form, as demonstrated in a 1D column by Atchley et al., (2016-section 4.2) and by Abolt et al., (2020) for wide troughs. This is because the timing of phase change during freeze up when snow is building can cause wet areas to stay warm throughout the winter, especially as the amount of water increases because it then requires a lot more energy loss to cross the freeze curtain. The difference with the study presented here is: 1) vary little surface inundation occurs due the surface runoff boundary condition and the assumed energy equilibrium at the downhill domain boundary condition. This assumption may not capture the thermal influence of the saturated

condition beyond the boundary of the domain. And 2) very little snow accumulation occurs, which would otherwise insulate the more saturated areas during freeze up. Given that the simulations with added precipitation showed an overall decrease in ALT, this work might suggest that increased evaporative cooling affect may outweigh the increased energy loss necessary to cross the freeze curtain. However, given the larger thermal hydrology work in literature, it would be beneficial discuss these tradeoffs as well as discuss how influential an appreciable snowpack may change the results. i.e there could be a combined warming in the dry areas (little evaporative cooling) and persistent warm winter conditions in wet areas from insulative snowpacks.

Abolt, C.J., Young, M.H., Atchley, A.L., Harp, D.R. and Coon, E.T., 2020. Feedbacks between surface deformation and permafrost degradation in ice wedge polygons, Arctic Coastal Plain, Alaska. Journal of Geophysical Research: Earth Surface, 125(3), p.e2019JF005349.

Atchley, A.L., Coon, E.T., Painter, S.L., Harp, D.R. and Wilson, C.J., 2016. Influences and interactions of inundation, peat, and snow on active layer thickness. Geophysical Research Letters, 43(10), pp.5116-5123.

Response: Thank you for this comment. As you mention, the formation of a potential talik is not so relevant for this study site since water does not start pooling above ground, even in the cases with twice the amount of precipitation, and snow cover is overall very shallow, due to the very dry climate in Adventdalen (the hydro-meteorological forcing conditions are taken from site data). However, in general ponding could occur, and surface water formation would in turn likely trigger talik formation. We have attempted to discuss various effects mentioned in your comment in the Outlook section (section 3.7) of the revised manuscript (previously "Further implications"), where we have adjusted the text and incorporated citations to Abolt et al. 2020 and Atchley et al. 2016 in this part.

L465-475

**Minor comments:**

1. L58-59:  In this case the benefit of numerical modeling probably has less to do with the remoteness of the study area, and more to do with a characterized sensitivity study as well as being able to dissect the energy fluxes across the full domain, something that would take tons of sensors to do in the field.

Response: Thank you for this comment. Indeed, we agree and have adjusted the text accordingly.

L57-60

2. L72: Change 'Tho' to 'To'

Response: Corrected.

L70

3. L89:  (Painter, 2011) is outside of either sentence, I think it goes with the previous one.

Response: Corrected.

L91 and L95

4. L118: Omit one of the 'a steep case'.

Response: Corrected.

5. L147-148: Why would you want to maintain a constant (same) snow accumulation across the hillslope domain, and is that realistic? This is likely to have a strong effect of simulations results.

Response: In this study we wish to investigate the effects of slope inclination on the hydrothermal conditions of the active layer. To enable clear and consistent comparisons we keep the boundary conditions and the hydrothermal forcing identical between the different cases, and enabling snow redistribution would potentially obfuscate comparisons. Also, since the site conditions are dry with very little snow precipitation, the effect of snow insulation and melt are minimal. However, adding snow redistribution would certainly be interesting for sites with more snowfall, as the lower end of the slope could then represent a snow pit/trap, and under those conditions much of the snow could end up on the lowermost part, and would then likely influence subsurface conditions locally. We provide additional clarification and justification of our model design in section. 2.2 "Simulation configurations".

6. L153-161: Note that these CV locations are right at the domain boundary, and therefore subject to any edge effects of the model domain. The assumptions and implications of being on these domain boundaries needs to be discussed in more detail. This might be especially problematic with the downhill CV, given that the no flow (energy) boundary condition assumes equilibrium with a larger body of water or saturated area.

Response: Thank you for the comment. We address this question in more detail in our response to Reviewer 1 (please see General comment 1 of Reviewer 1). This has led to a slightly modified model domain, which more clearly represents the valley base and allows for energy and water fluxes out of the slope. The location of the CVs on the outermost boundaries of the domain was intentionally chosen to capture the most extreme values (drying on uphill side, and accumulation on downhill side). Now with the revised model domain, the downhill side CV is not on the right-most boundary anymore. Aslo, potential boundary effects are evaluated (see response to General Comment 1 of Reviewer 1). The intention of the placement of the CVs has been clarified in the main text, in the Section 2.2.

7. L207-208: This sentence needs to be more specific. What do you mean by inversion of temperature differences? Differences between uphill and downhill observations? Or differences between sloped and flat? Also, what are the different processes responsible here?

Response: We restructured the text and removed the paragraph that contains this unclear formulation.

8. Figure 3 needs more explanation in the caption. Hard to interpret it at a glance.

Response: We have made attempts to clarify this figure and caption text; we extended the caption text with an explanation indicating what positive and negative values mean in each of the panels. Further, we added the original time series from which the differences are derived, and placed this in the supplementary information (Fig. S3).

9. L215: "The upper three panels in each figure…". I assume you are talking about Figure 4 here, but it is not clear. I suggest phrasing this as, "The upper three plots in each panel…" as 'figure' refers to figure 1 through 9 in the paper, 'panel can refer to a and b, and plot is the sub plot of each panel.

Response: Thank you for the suggestion. We adjusted text according to the comment.

L238-240

10. Figure 7: I like this figure, or at least what it is attempting to convey. However, I think it could be improved, or perhaps simplified. Is the story in the time series of the flux? Is it necessary to show the whole time series? If not, I would suggest just showing the cumulative flux, or maybe the difference of the fluxes between the cases. That may help with the scale issue. Additionally, the location of the representative volumes in relation to the fluxes going in and out is confusing, i.e, the lateral energy flux going in the downhill volume (positioned right in the domain) is found left in the figure. This means the reader (me) has to mentally flip the image.

Response: Thank you for the comment on this admittedly complex figure. We tried to simplify the figure according to your very useful feedback. Below you can find the figure with the cumulative sum (Fig. 2 in this letter), however this unfortunately leads to a significant loss of information in terms of direction of fluxes (specially for the vertical fluxes). The new final figure includes adjustments to the moving average (90 day window) and placing of the uphill and downhill panels as well as an adjustment to the titles (Fig. 3 in this letter and Figs. 7 and 8 in the main text). We aimed for the best compromise between correctness and simplicity. This should improve readability.

[Figure]

*Figure 2: Advection and diffusion of heat through the control volumes (cumulative sum) (not used)*

[Figure]

*Figure 3: Advection and diffusion of heat through the control volumes (90 day moving average) (used in revised manuscript as Figures 7 and 8)*

Figure 7

11. Figure 8:  These are pretty small fluxes.  Is this figure necessary?  I think this paper as a whole makes a decent argument that groundwater dynamics are important in determining the thermal state of the hillslope, even if advective fluxes only play a small role.  In other words, the influence of groundwater dynamics happens in other processes.  I would suggest focusing as much as possible on those processes rather than advection.

Response: Thank you for this remark. Even though the values are fairly small, especially vertical advection plays a big role in the differences between up- and downhill (namely evaporation and infiltration). As described in the text, the uphill section is mostly dominated by infiltration rather than evaporation, which on the other hand dominates in the downhill section.

No changes

12. L462:  "as compared to the flat case (0.75cm)." is confusing.  Does the flat case change in the simulations?  I thought the flat case provided a reference datum and therefore should be 0.

Response: Thank you for catching this typo. This sentence was supposed to refer to the active layer depth of the flat case in absolute values and should say 0.75m instead of cm. So the baseline active layer depth was 75cm and was increased in the steep case by 5cm (as stated in the sentence) to 80cm. Note that the active layer depth has changed to max. 1.03m  in our new simulations, mainly because we have corrected the thermal conductivity values (please see our response to Comment 10 by Reviewer 1). Thus, the values have been updated in the revised text. For completeness, we have added the maximum active layer depth for the medium case as well, so all three cases have their respective absolute maximum active layer depth.

L481 and L486-489

13. Figure 9:  "The sign convention adopted is positive values represent heat entering the CV and negative values leaving the CV."  This would mean that I should see the evaporative cooling affect?  Correct?  Heat and mass leaving the downslope CV during summer?

Response: Your assumption is correct. However, we are only showing the lateral mass flux here to explain the relationship between advective heat transfer and mass flux. To see mass fluxes in vertical direction, please refer

to Figure S7 (in the supplementary information), which includes mass fluxes in vertical direction and shows the influence of evaporation and infiltration.

No changes

14. Section 3.6: This section tests the effect of overall precipitation and demonstrates the overall effect of increased evaporative cooling – at least I think that is the purpose. However, as written it seems to play only a minor role in the manuscript (the text devoted to this seems like an after thought), and there is no figure associated with this text that actually illustrates the point other than those in the supplementary information. It would be better to have a figure in this text.

Response: Thank you for these suggestions. We revisited the section and added a thaw-depth-plot with both scenarios (S0R0 and S2R2) analogous to Figure 5a in the main text (panel b of Fig. 5 in the main text has been removed). The new plot (see Fig. 4 in this letter) is now part of the main text (Fig. 10 in the main text) and is removed from the supplementary material. We have also improved and simplified the text around the sensitivity study.

[Figure]

Figure 4: Spatial mean active layer depth for scenarios S0R0 (a) and S2R2 (b). The thaw depth indicates the transect-average thaw depth of each case (steep, medium, flat)

Section 3.6

15. L424-L432: This paragraph provides a good summery of what was found. It seems like it would fit better in the conclusion section.

Response: Thank you for the suggestion. We agree and have moved this part of the discussion to the conclusions section.

L477-480

16. L458-46: "This downhill cooling effect is up to about 1.2 C for steep (22°) and 0.6 C for medium (11°) inclinations across a lateral distance of 50m representative for valleys in Adventdalen, Svalbard." Seems to me that the bigger story here is not the increased downhill cooling caused by the evaporative flux, but the

increased uphill warming presumably caused by the lack of an evaporative flux because overall the entire domain is warmer than the flat case.

Response: We agree that the focus of the study was too much shifted towards the downslope cooling effect while the uphill warming has fallen a bit too short. We revised the manuscript accordingly to put the uphill warming more into the focus and highlight that the warming actually causes the active layer to be deeper. Furthermore, the downhill cooling effect is moderated with the revised model domain.

Several occurrences in the text, e.g., L490-499

17. The conclusion section is not very impactful.  The paragraph in L424-432 seems to be a much better conclusion paragraph. Also, after reading the paper several times, I am confused as to why the entire sloped domains are over all warmer than flat domains?  The focus appears to be more on the evaporative cooling effect, yet the domains are overall warmer unless precipitation is increased.

Response: Thank you for the tip. As mentioned above, we have merged  the paragraph from L424-432 to the conclusions section. Indeed, the summer warming of the sloped cases compared to the flat case is mainly attributed to additional heat through infiltration and overall lower moisture content causing a lower heat capacity and reduced evaporation. We now also address heat capacity and more clearly emphasize the overall warming effect on the uphill section of the domain (see general comment 1 above).

Section 4

---

## Author Response (AR2)

**Response letter to Referee #3**

We would like to thank the reviewer for the time they have taken to offer insightful and constructive comments to our manuscript. We have addressed the comments in our responses to the reviewer below. This has led to several improvements and clarifications to our study. In particular, the reviewer was concerned about the lack of representation of direct model output and the numerical and physical reliability of our results.

To improve the presentation regarding model outputs, we have further extended the supplementary information with figures depicting direct model output. These are aimed for the interested readers and to further support and clarify our findings. We have also added a new conceptual diagram in the main text to aid the discussion on saturation, thermal conductivity and heat capacity. The numerical accuracy of simulations is ensured by assigning stringent numerical convergence criteria, monitoring output, and checking results are numerically stable. Therefore we expect the model outcomes to be robust and valid solutions of the system of equations. The model applicability and consistency with real-world conditions is achieved by using realistic boundary conditions and careful model design, including choice of parameter values and hillslope inclinations consistent with observations. Further, the model output in terms of thaw depth and active layer thickness is compared against measurements in Adventdalen, thereby attaining model confirmation (Oreskes et al. 1994) against field measurements.

With these changes and clarifications we believe the manuscript has been significantly improved and is highly relevant for the readership of the Cryosphere.

Our responses are written in blue font. Figures shown in this response letter are referred to as Figure R#. Otherwise, figure references refer to the main text.

Line numbers in the **revised manuscript** that contain changes are given in red font.

**General comments:**

The manuscript reports a numerical modeling study aiming to better understand lateral groundwater flow influence on temperature and heat transport in active layers. The study is well motivated and the presentation is well organized. I have some concerns on how the modeling is done and consequently how meaningful the interpretation of the model results is. Below are the specifics.

1. The model domain has a relatively small lateral dimension (~ 50 m), "a very short slope" as the authors recognize but justify it by needing to provide a trade off between resolution and computer time (Line 165-166). My concern is that the trade off may sacrifice the appropriateness of the model representation. Given the focus of this study is on processes in the lateral direction, the small lateral dimension casts much doubt on the validity of the interpretation of model results.

Response: Although the simulated hillslopes are relatively short, they are valid hillslope representations, certainly for small catchments. The results we observe are based on numerical representation of physical processes, as simulated by the model, and therefore represent relevant findings of this complex flow system. As mentioned in the main text, due to the model complexity a high vertical mesh resolution is required in the uppermost parts of the domain, in particular where the active layer is located, which greatly increases the computational effort. Consideration of larger systems is certainly of interest and deserves investigation, but is beyond the scope of our current study.

Furthermore, even if these results are in a strict sense limited to the distance considered, we consider possible effects of larger slope systems/catchments in the outlook section (section 3.7). Also, we include a sensitivity analysis of two additional wetness scenarios, which simulate dryer and wetter conditions than the current hydro-climatic conditions (section 3.6). We adjusted the sentence mentioned in the comment slightly for clarification (previously L165-166).

Changes: L168-170: The domain size is chosen to represent a generic hillslope that provides a reasonable trade off between model resolution and computational effort.

2. For a small model domain like this, boundary conditions could strongly dictate model outcomes. Much of the model interpretation is based on two control volumes near the left and right boundaries where no-flow (heat and water) conditions are applied. While a no-flow condition across the right boundary may be justified on the ground of valley symmetry, the no-flow condition for the left boundary, a short distance from the valley floor cannot be easily justified. The boundary condition on model top is vague without specific details. A slight change in these boundary conditions would likely lead to different model outcome. Ultimately the modeling here is to solve a boundary value problem.

Response: The control volumes are intentionally placed to monitor flows on the extreme ends of the slope, and to encompass the part of the active layer for which the most significant heat- and water flow dynamics occur. The no-flow boundary condition on the left side represents a water divide. This is chosen to avoid arbitrary inward mass flux, which would indeed obfuscate results. The boundary condition on the surface of the domain is a surface energy balance using hydro-meteorological inputs. This is presented in the main text (Section 2; Data and method) and supplementary information Figure S1. The surface energy balance model is described in Atchley et al. 2015. As such, the model is driven by the hydro-meteorological input, which is derived from weather station measurements in Adventdalen, Svalbard. Therefore, the model is not driven by an arbitrary set of boundary conditions and the results are robust, and only sensitive to the imposed hydro-meteorological input, which again, comes for real-world observations. This is one of the powerful features of the approach undertaken by the ATS model.

Furthermore, as mentioned above in response to comment 1, we also investigate the sensitivity of the system with respect to the current hydro-climatic conditions by considering two additional hydro-meteorological scenarios derived from the original weather station input, one corresponding to dry conditions and another one to wetter conditions (S0R0 and S2R2, Section 3.6; Impact of changes in precipitation).

The "short distance from the valley floor" (we believe this is referring to the distance between the foot of the slope and the right boundary) was chosen to be 16m long (8 columns). We have investigated the effect of a shorter "valley bottom" of 4m and found no significant differences between those two scenarios and therefore conclude that 16m in the valley bottom is more than suitable to avoid any boundary effects on that edge. This information is not included in the manuscript but was previously provided in our response to Referee #1, where we state how we approached the valley bottom geometry.

No changes

3. On model initialization and spin-up (section 2.2.2), it is puzzling why the authors used 33 1D column and then map the result to the 2D model, instead of using the actual 2D domain to initialize the model. A common way to initialize models is to use time-averaged conditions to let the model spin up to a quasi-steady background state and use the quasi-steady state as the initial condition for transient runs. The unusual approach used in this study may pose additional problem for the no-flow conditions across the two lateral boundaries.

Response: The initialization procedure adopted is well-established and consistent with previous efforts for modeling Arctic/permafrost hydrogeological systems (e.g. Painter et al. 2016, Jafarov et al. 2018, Jan and Painter 2020). The procedure used, described in section 2.2.2, is necessary to obtain a physically consistent, periodically stable annual flow system undergoing freeze-thaw. Note the initialization procedure includes a spin-up also for the full 2D domain, after the mapping of the 1D column has been performed, which allows for stabilization of lateral processes (step 4 in section 2.2.2).

Our intention is to investigate an annually periodic system consistent with recent climate conditions and therefore longer term transient runs are intentionally not performed. The periodic steady-state is ensured by using the last (100th) year of the final stage spin-up. In our evaluations we have determined 100 years for the final stage spin-up to be more than sufficient to obtain annually stable/consistent conditions.

The lateral no-flow boundaries are intentionally assigned as such as they represent symmetry boundaries. They do not pose a problem for the flow field because the top surface of the domain allows for both recharge and discharge to occur and the water table during unfrozen conditions is a free-surface.

No changes

4. Model result presentation needs much improvement.

4.1 Presenting temperature difference (Figure 4) is difficult to make sense. Direct modeled temperature outputs for different scenarios need to be presented (even in supplement). The difference of two wrong sets of data may look reasonable. Please excuse my bluntness, I do not mean to say that the model results here were wrong, but just to say that the difference may not tell the whole story.

Response: We find temperature differences to be the most convenient and clear way to present and compare results. For the interested reader, the direct modeled temperature

outputs are indeed available in the supplementary information (Figure S3) as mentioned in L215-116 in the old version of the manuscript (now L218-219).

No changes

4.2 The temperature time series or whatever condition is applied on the top boundary need to be added to the top of Figure 3 for readers to make better sense of the modeled temperature results at three depths. For example, I am puzzled by the wiggles in all of the modeled temperature time series. Is it because of the temperature variations in the boundary condition propagating down or because of potential numerical errors? It is also puzzling why temperatures at 0.5 m experienced the most dramatic changes while deeper and shallower temperatures are more subdued.

Response: A surface energy balance is used for the top surface of the model domain, which includes several input variables in addition to temperature. For completeness, these are presented in Figure S1 in the supplementary information. Due to the complexity of surficial processes, including the SEB and the fact that hydro-meteorological data with considerable daily variation is used as input, the surface signals propagate into the subsurface causing day-to-day changes in subsurface state variables. These are a result of system behavior and not numerical artifacts. We added a sentence to explain that the forcing dataset has not been smoothed before applying it as a boundary condition in the main text as well as in the figure caption of Figure 3)

We address the observation that temperature differences at 0.5m depth are the most considerable (L225-226 of the previous version of the manuscript, now L228-229) and it can also be seen in Figure 4, which presents temperature differences in the active layer and the upper permafrost. With Figure S7, we explain that heat capacity has a major influence on temperature differences as described in L320-321 (previous version of the manuscript, now L327-329)

Changes: L119-120:Apart from the redistribution of precipitation, the meteorological data has not been smoothed.

4.3 The unsmooth curve for the flat slope scenario (Figure 5) also is puzzling. If there are any numerical issues with this base case flat slope scenario , then the other two cases comparing with the base case may be problematic.

Response: The incremental changes in thaw depth in Figure 5 are a reflection of the mesh resolution of the model domain in the active layer (5cm depth, cf. Figure S2) and because we chose to present direct modeled results without smoothing or interpolation. We define thaw depth as the depth for which the entire mesh volume is at or above 0 degC, therefore the thaw depth propagation over time has an incremental nature. Note also that the thaw depth shown in Figure 5 is a spatial average over the entire transect length, which varies slightly for the inclined cases but not for the flat case. Therefore, the sloped cases appear to have a smoother propagation over time.

Changes: Figure 5 has been moved  from the main text to the supplementary information following a suggestion by Referee #4

4.4 The discussion on saturation, thermal conductivity and heat capacity (section 3.3) is laborious and stressful to read. A conceptual illustration may aid the discussion. My bigger concern is that if the numerical model results were not proper, then discuss would be strenuous.

Response: We have made efforts to improve this part of the presentation, both text and by adding a new conceptual figure as suggested (Figure 6 of the revised manuscript and Figure R1 below), which we believe helps clarify the discussion and analysis of this section.

[Figure]

Figure R1: Conceptual diagram of the effects of saturation on ground temperatures in the active layer in summer time. The arrows indicate if the quantity is increased (up, dark) or decreased (down, light).

Changes: Section 3.3 and Figure 7.

5 The authors need to provide a big picture about what the groundwater flow field is like, lateral and vertical, given the intention of this paper is to look at the role of lateral groundwater flow. Direct model output in terms of groundwater head (or pore pressure) field and water flow velocity field in the main text or in supplement would go a long way to provide key model results necessary for readers to comprehend how the physical process of lateral flow influences heat transport in active layers. One piece of information may be the mass flux (Figure 9), but then why the dip in mass flux (meaning recharge?) in November in downhill locations?

Response: Our intention is to show condensed information of the model output, such as temperature differences in the main text to enable the analysis. However, direct model output can be of interest for highlighting technical details, and for this reason we have added a selection of plots in the supplementary information. Due to the vast number of possible depictions, including multiple simulations each with several output variables and with transient dynamics occurring over the year, it is not feasible to show a full set of plots consistent in both space and time. Therefore, we chose a selection of variables for specific points in time focusing on the upper part of the domain (upper 1.2m) analogous to Figure 4 in the main text. This direct model output includes temperature, liquid-, ice-, and gas

saturation. This has been added to the supplementary information (Figure S5 and Figure R2 below).

We have omitted representations of the pressure field and darcy velocity field because they are not informative for this analysis; an example of the pressure field with a restricted pressure range can be seen below (Figure R3). Snapshots of darcy velocity vector fields are not informative in our case because of the transient nature of the simulations.

Figure 9 depicts lateral fluxes of heat and water flow. The negative values occurring in November in Fig 9b,d indicate fluxes leaving the downhill CV across its vertical face (at x=48 m). This is an effect of lateral cryosuction as well as two-sided freezing and pressure differences in the domain.We discuss and elaborate on this in Section 3.5

[Figure]

*Figure R2: Representation of liquid- (rows 1 and 2), ice- (rows 3 and 4), and gas saturation (rows 5 and 6) on summer day (July 20) and a winter day (November 18) throughout the transect (representation of the upper 1.2m of the model domain across the 50m slope transect). Red colors represent low saturation, blue colors high saturation.*

[Figure]

*Figure R3: Representation of pressure on summer day (July 20) and a winter day (November 18) throughout the transect (representation of the upper 1.2m of the model domain across the 50m slope transect). Red colors represent high pressure, blue colors low pressure. The lower range of pressure has been restricted to 0.5 MPa for better visualization.*

Changes: L404-410: Note also that during freeze-up (November) in the downhill CVs, there are negative values for mass flux (Fig. 9b,d), indicating moisture is leaving the CV in the uphill direction, which we attribute to two-sided freezing and lateral cryosuction. While the active layer starts freezing from above, it also freezes from below, causing high water pressure in the remaining space occupied by liquid water. Due to the temperature distribution in the slope and valley bottom, the only direction the water can be squeezed out towards is uphill. Even though this effect might be overemphasized in a 2D domain, it is a physical based effect unique to permafrost landscapes. Additionally, unfrozen water in the downhill side of the domain can migrate towards the freezing front approaching from the uphill side (lateral cryosuction).

6 The relative magnitude/significance of heat conduction versus heat advection by groundwater is unclear. After all, this study aimed to examine the effects of groundwater on temperature, basically advective heat transport. Conduction and advection were presented in separate figures 7 and 8 and the axis scales differ. It is not easy to compare them, for example, a basic question is what the relative percentages of energy transport by conduction versus advection are. An additional figure to show the system energy balance would be helpful.

Response: The advective and diffusive heat transport are intentionally depicted in separate figures for clarity of presentation. The ratio is also interesting and we have calculated the Péclet number for each of the faces of the CVs as well as the entire model domain and

included this as a new figure in the supplementary information (Fig. S8 and Figure R4 below) to avoid excessive detail in the main text.

[Figure]

*Figure R4: Daily ratio between advective and diffusive energy flux on each of the faces of the a uphill CV, b downhill CV and c the entire CV. Solid lines represent values for the steep case, dashed lines represent the medium case, while colors indicate the different faces of the CV. Dashed horizontal lines in a and b indicate the value of 1, where the advective energy flux becomes more pronounced than the diffusive energy flux. Note that there is no such line in c, as the Péclet number over the total CV is very small.*

Changes: Figure S7

7 Model result interpretation may be questionable. No attempt is made to model calibration. At the very least, a first order check of the model results with any field observations would be necessary to convince readers that the model results make sense. For example, one model result is that warmer temperature in uphill and cooler temperature in downhill slopes (Section 3.1, Conclusion i). My intuition seems to be the opposite. The explanation provided (evaporative cooling) is quite strenuous and unconvincing. Any broad observational data may support such model results? Similarly, any broad observations that suggest "steep slope develops deeper thawing front" (Line 255)?

Response: The purpose of this study is to investigate effects of hillslope inclination using realistic conditions; a site-specific study with calibration, inverse modelling or parameter estimation is neither intended nor necessary. Note that the model is a physically-based model adopting conservation equations for energy, mass and momentum and in addition to using realistic physical parameters (Table 1) and hillslope inclinations (Table S1), site-specific hydro-meteorological data are derived and used as input for the surface energy balance and top surface boundary condition to ensure realistic weather variability conditions, thereby achieving relevant and realistic simulation scenarios as needed for this study.

Available field observations are used to attain confirmation (Oreskes et al. 1994) of the applicability of the model, which includes measured active layer thickness in Adventdalen, Svalbard (Strand et al. 2020, Schuh et al., 2017), where our simulated ALT are consistent with those measured ALT. This information is included in Section 2.1 "Field data".

To clarify the model consistency with field measurements, we have added a sentence in Section 3.2, where we analyse the progressing thaw depth in each of the cases and explain that those simulated values are consistent with ALT measurements in Adventdalen, with citation to the studies mentioned above. This shows that the model is indeed capable of simulating the hydrothermal state of the active layer very well, in fact remarkably well considering it is a forward model.

Our results indicate a downhill cooling effect which may indeed be contrary to initial assumption or intuition, and we are very excited about this important discovery. In our study we carefully and meticulously analyse the phenomenon and provide physically-based, mechanistic explanations of the effect. As such, our study is robust and of great significance to the cryosphere community and warrants prompt distribution.

No changes

In conclusion, I like this study but feel quite uncomfortable about the modeling approach and consequently the results. More direct model results must be presented before readers can assess the interpretation of model results. More bluntly, without showing those direct model output, I am not confident that the results are good. (BTW, my research has involved numerical modeling on water and heat transport in porous media for decades.)

Response: We appreciate and greatly value the careful scrutiny and attention to detail. We are confident we have thoroughly addressed these and made all necessary clarifications and amendments to our presentation.

 **References in this letter**
Atchley, A. L., Painter, S. L., Harp, D. R., Coon, E. T., Wilson, C. J., Liljedahl, A. K., and Romanovsky, V. E.: Using field observations to inform thermal hydrology models of permafrost dynamics with ATS (v0.83), Geoscientific Model Development, 8, 2701–2722, https://doi.org/10.5194/gmd-8-2701-2015, https://www.geosci-model-dev.net/8/2701/2015/, 2015.

Jafarov, E. E., Coon, E. T., Harp, D. R., Wilson, C. J., Painter, S. L., Atchley, A. L., and Romanovsky, V. E.: Modeling the role of preferential snow accumulation in through talik development and hillslope groundwater flow in a transitional permafrost landscape, Environmental Research Letters, 13, 105 006, https://doi.org/10.1088/1748-9326/aadd30, http://stacks.iop.org/1748-9326/13/i=10/a=105006?key=crossref.ea8d38a9a41cbb12 0144acdd5d1d4d37, 2018.

Jan, A. and Painter, S. L.: Permafrost thermal conditions are sensitive to shifts in snow timing, Environmental Research Letters, 15, 084 026,

https://doi.org/10.1088/1748-9326/ab8ec4,
https://iopscience.iop.org/article/10.1088/1748-9326/ab8ec4, 2020.

Oreskes, N., Shrader-Frechette, K., Belitz, K., 1994. Verification, Validation, and
Confirmation of Numerical Models in the Earth Sciences. Science 263, 641–646.
https://doi.org/10.1126/science.263.5147.641

Painter, S. L., Coon, E. T., Atchley, A. L., Berndt, M., Garimella, R., Moulton, J. D., Svyatskiy,
D., and Wilson, C. J.: Integrated surface/subsurface permafrost thermal hydrology:
Model formulation and proof-of-concept simulations, Water Resources Research, 52,
6062–6077, https://doi.org/10.1002/2015WR018427,
http://doi.wiley.com/10.1002/2015WR018427, 2016.

Schuh, C., Frampton, A., and Christiansen, H. H.: Soil moisture redistribution and its effect
on inter-annual active layer temperature and thickness variations in a dry loess
terrace in Adventdalen, Svalbard, The Cryosphere, 11, 635–651,
https://doi.org/10.5194/tc-11-635- 2017,
https://www.the-cryosphere.net/11/635/2017/, 2017.

Strand, S. M., Christiansen, H. H., Johansson, M., Åkerman, J., and Humlum, O.: Active
layer thickening and controls on interannual variability in the Nordic Arctic compared
to the circum-Arctic, Permafrost and Periglacial Processes, p. ppp.2088,
https://doi.org/10.1002/ppp.2088,
https://onlinelibrary.wiley.com/doi/10.1002/ppp.2088, 2020

Response letter to Referee #4

We would like to thank the reviewer for the time they have taken to offer insightful and constructive comments to our manuscript. We have addressed the comments in our responses to the reviewer below. This has led to several improvements and clarifications to our study. In particular, we improved the visualization of some figures and included a new conceptual figure to simplify the interpretation of our results. We further worked on text passages, which were unclear and improved readability.

With these changes we believe the manuscript has been further improved.

Our responses are written in blue font. Figures shown in this response letter are referred to as  Figure R#. Otherwise, figure references refer to the main text.

Line numbers in the **revised manuscript** that contain changes are given in red font.

This manuscript addresses a novel topic using a robust simulator of surface and subsurface cryohydrological processes and will be an important contribution to the cryosphere literature. The authors thoroughly address reviewer comments and have greatly improved the manuscript. However, the results/discussion section is still difficult to digest despite text revisions. I suggest separating the results and discussion section to more clearly convey the results and more comprehensively discuss the findings. If keeping the results and discussion section merged, I suggest further revision of the text (some suggestions included below) as well as inclusion of a conceptual diagram that visually displays the results. One option is to reorganize the section to enhance clarity, perhaps based on mechanism or question addressed (L69-72).

Specific suggestions:

1. L32-35: revise sentence for clarity. Perhaps "Further, higher moisture abundance in the active layer regulates the decomposition of organic carbon, can affect infrastructure built on the fragile frozen ground, and can change the thermal properties of the permafrost."

Response: Changed sentence according to the suggestion by the reviewer

Changes: L32-34 Further, higher moisture abundance in the active layer regulates the decomposition of organic carbon (e.g., McGuire et al., 2009; Koven et al., 2011), can affect infrastructure built on the fragile frozen ground (e.g., de Grandpré et al., 2012), and can change the thermal properties of the permafrost (e.g., Schuh et al., 2017).

2. L59: Change untangling to untangle.

Response: Changed

3. L142: Please extend the sentence to elaborate on why this is important for the choice of boundary conditions in the model domain. It is not clear from the sentence or paragraph as it is now.

Response: Extended the sentence and added examples for potential boundary conditions.

Changes: L143-146: This is important for the choice of boundary conditions in the model domain, which regulates the water flux out of the domain. Potential boundary conditions for this set-up could be either a closed boundary (no outflow) an open boundary (outflow through the surface and subsurface) or a constant head boundary, which would indicate a persistent river and allow for groundwater discharge into the river.

4. L153-154: Please add more detail with respect to the cell thickness. What is the maximum cell thickness?

Response: Added maximum cell thickness information in brackets (~1.5m)

Changes: L157-158: With increasing depth, cell thickness gradually increases (up to max. ~1.5m cell thickness).

5. L229: Remove comma after slopes.

Response: Removed

6. L231: Remove 'or'.

Response: Removed 'are' in L230 instead (old manuscript) and changed 'as' to 'to' ('or' is correct).

Changes: L233-234: Deeper layers have similar temperatures to the flat case or are even warmer.

7. L263: Perhaps the authors can add a few sentences discussing the patterns seen in the winter panels.

Response: We have added an explanation of the patterns in the December 7 snapshot seen in Figure 4. The new sentences are placed after L245 (of the previous version of the manuscript) to better fit in with the flow of the text.

Changes: L257-260: The patterns seen in both December 7 plots (red patches between -0.2 and -1.2m) are consequences of the timing of freezing in the slopes. While the flat cases freezes uniformly, the active layer slopes freezes faster uphill and slower downhill, causing those temperature differences.

8. L282: Remove "does" and replace "play" with "plays".

Response: Changed

9. Figure 4: Is it possible to add the O degree C isotherm for the flat case to each plot? This addition could also take the place of Figure 5. Given that you discuss differences in the thaw depth between the steep/medium and flat case in the text, it would be beneficial to see that

difference in the figure. Perhaps the flat isotherm could have a different color and line pattern.

Response: We appreciate the idea of including the flat 0°C isotherm into Figure 4 and changed Figure 4 accordingly (see Figure R1 below). We also agree that Figure 5 is partly redundant with this figure and moved it to the supplement.

[Figure]

Figure R1: Temperature difference between **a** the steep and the flat case and **b** the medium and flat case at six selected dates highlighting thaw, summer, freeze-up and winter. Red colors indicate warmer temperatures in the hillslope cases than in the flat case, blue colors indicate cooler temperatures (note the color scale differs between summer and winter comparisons). The black dashed lines indicate the 0°C isotherm(s) in the corresponding hillslope cases (steep and medium) at the respective dates. The 0°C isotherm lines of the flat case are represented by dotted lines. During

*freeze-up, it can be seen that two-sided freezing occurs. (For clarity, only the upper 1.2m of the simulation domain is shown.)*

Changes: Figure 4 in the main text and Figure S4 in the supplementary information

10. Figure 5: If keeping this figure in the manuscript, please consider including three line styles in addition to three colors to help see different in lines.

Response: Added line styles in addition to the colors and moved previous Figure 5 (now Figure S4 and Figure R2 below) to the supplementary information and the text has been adjusted accordingly.

[Figure]

*Figure R2: Representation of thaw depth compared between the steep (blue), medium (cyan) and flat case (yellow) as daily, spatially averaged thaw depth (averaged over a 5-day window) from May to December in the last year of the simulation. Note that thaw depth is defined as cells within the model domain that exceed 0°C.*

Changes: Figure S4 and L270-273: The spatial mean active layer depth in the deep slope on the date of maximum active layer depth is 1.03 m (min.:1.03 m, max.: 1.03 m along the transect). The medium slope exhibits a smaller uphill warming than the steep slope resulting in a spatial mean active layer depth on the date of maximum active layer depth of 0.986 m (min.:0.975 m, max.:1.030 m along the transect), which is only slightly deeper than in the flat case (0.975 m).

11. L289: Please provide more detail to clarify your reasoning for a more saturated downhill side on the medium slope.

Response: The main reason for higher liquid saturation in summer in the downhill side of the slope is the geometry of our slopes. As can be seen in Figure R3 below, gravity driven water flow causes the fully saturated cells in the steep slope to be deeper (40cm depth) than in the medium slope (35cm depth). On the other hand, the first column in the valley bottom (which we do not discuss in the manuscript) is saturated up until 25cm below the ground surface in the steep case and only until 30cm below the ground surface in the medium case.

[Figure]

*Figure R3: Subsurface water level (red colors) in the foothill of the steep (left) and medium (right) slope in summer. The depth below the ground surface at which the soil is fully saturated is given in cm.*

We changed the text in the revised manuscript and removed this superfluous comment because it diverts from the main message of the section. As pointed out by the reviewers, this section (Section 3.3) is already very complex and therefore we hope to make it easier for the reader to follow our line of thought.

Changes: Removed L287-298 in the old manuscript

12. L289-292: I suggest removing the '/' and separate this sentence into two sentences for clarity.

Response: We restructured the sentence for clarity

Changes: L289-296: Due to gravitational flow of water during the warm period, moisture is drained from the uphill side and accumulates on the downhill side, reducing liquid saturation uphill and increasing it downhill when compared against the flat reference case which is not subject to lateral flow (Fig. 5, first column). This leads to differences in ice saturation during the frozen period (Fig. 5, second column), specifically reduced ice saturation uphill and increased downhill. Consequently, the uphill side of the sloped cases experience increased air saturation (Fig. 5, third column), which yields a considerably lower effective thermal conductivity during winter and slightly lower effective thermal conductivity during summer (Fig. 5, fourth column). Similarly, the downhill side has reduced air saturation (Fig. 5, third column), yielding greater effective thermal conductivity; considerably greater during winter and slightly greater during summer (Fig. 5, fourth column).

13. L300: I suggest rephrasing this topic sentence. Perhaps modify sentence 1 and merge with sentence two.

Response: This sentence has been improved by reorganizing the beginning of this paragraph.

Changes: L306-308: Recall the previous discussion on temperature differences between the sloped and flat cases (Section 3.2). The uphill sides of the sloped domains (Fig. 3c,e) are slightly drier at depths 0.2m, 0.5m and 1m, both for summer with less liquid saturation, and winter with less ice saturation (Fig. 6, first and second columns, respectively).

14. L313: I suggest rephrasing this topic sentence. More specific topic sentences will help clarify this section, as it is long and detailed and can be difficult to follow. Additionally, this is one place where a conceptual diagram to support the discussion would be useful.

Response: Parts of this section have been rephrased. A new conceptual figure added (Figure 6 in the manuscript and Figure R4 below).

[Figure]

*Figure R4: Conceptual diagram of the effects of saturation on ground temperatures in the active layer in summer time. The arrows indicate if the quantity is increased (up, dark) or decreased (down, light).*

Changes: L333-336: In summary, moisture redistribution mainly causes differences in thermal conductivity and heat capacity between the uphill and downhill sections (Fig. 6). Thermal conductivity mainly affects energy transport by conduction, and heat capacity attenuates transport by storage. However, to fully understand the effects of energy transport on ground temperatures, a complete analysis of energy fluxes is needed, which is discussed in the next section.

15. L332: Add (CV) after control volume.

Response: Added

16. L349: Smaller than what? Please expand the sentence.

Response: Added missing information

Changes: L357-358: In the uphill CV (Fig. 7b, solid), the lateral heat diffusion is more than one order of magnitude smaller than in the downhill CV (-0.01–0.015 W m$^{-2}$) and heat is being lost in summer, but gained after freeze-up in winter.

17. L404: Perhaps the authors can include some numbers with this statement such as average temperature.

Response: We included a table (Table S4 in the revised supplementary information) with average upper domain (up to 1.2m depth) temperatures for the original scenario (equivalent to Table 2 in the main text) alongside active layer temperatures in the two precipitation scenarios. We also added numbers for the relative difference between the slopes and the flat case.

Changes: L417-421: Firstly, we find that both slopes and the flat case are notably warmer in the no-precipitation scenario (S0R0) and colder in the doubled precipitation (S2R2) scenario (Table S4 in the supplementary material). Relative temperature differences between the slopes and the flat case are generally in a similar range as in the original precipitation scenario. The steep slope in S2R2 is up to 0.7°C warmer than the flat case in summer and up to -0.6°C colder in winter. In S0R0, the steep slope is up to 0.3°C warmer than the flat case in summer and up to -0.2°C colder in winter.

18. Figure 10: Consider distinguishing lines with line style (dot, dash) as well as color.

Response: Added line styles in addition to the colors in Figure 10 in the main text.

[Figure]

*Figure R5: Representation of thaw depth compared between the steep (blue), medium (cyan) and flat case (yellow) as daily, spatially averaged thaw depth temporally averaged over a 5-day window from May to December in the last year of the simulation. Note that thaw depth is defined as cells within the model domain that exceed 0 °C.* **a** *shows the results for the S0R0 (dry) scenario, while* **b** *shows daily thaw depths for the S2R2 (wet) scenario.*

Changes: Figure 10

19. L416: Remove 'the' before 'both'. I suggest rephrasing the sentence for clarity.

Response: We have rephrased the sentence to make it more clear

L432-433: Overall, the scenarios show that a higher amount of recharge added through precipitation on the surface will decrease the ground temperatures in the slopes as well as in the flat case.

20. L440: Replace 'where' with 'were'.

Response: Corrected

21. L479: Remove the comma after 'both' and after 'conductivity'.

Response: Removed

---

## Author Response (AR3)

**Response letter to comments by the editor**

We would like to thank Ylva Sjöberg for her time to offer constructive comments to our manuscript.
Our responses are written in blue font.

Line numbers in the **revised manuscript** that contain changes are given in red font.

**General comment:**

Thank you for submitting a revised version of your manuscript (tc-2021-60) together with responses to the comments for reviewers #3 and #4. I find that you have addressed almost all concerns from the reviewers very well and that the manuscript is near a final publication in TC. However, I have a few minor comments remaining before publication and have therefore decided that minor revisions are needed on your manuscript. These should not require much work from you and I hope we can therefore soon see your very nice contribution to this topic published in TC.

**Specific comments:**

1. Section 2.1: clarification is needed about what data is in the end used to run the model (resolution, processing, length, variables). Specify that (if!) it is the locally observed daily values of air temperature, relative humidity, wind speed, and shortwave radiation that was used for simulations together with the processed (as described) precipitation data (averages?) for the full (?) years of 2013 to 2019. Also, that a set of spinup data was produced from this data (as described).

Response: We added more information about the forcing dataset used to run the model, such as resolution of the original data as well as of the final dataset, variable names and the length of the full forcing dataset. In the end of Section 2.1 we describe how we created the forcing data set for the simulations.

Changes: L109-129: The observational weather data to drive the model (hereinafter referred to as the forcing dataset) is derived from an automatic weather station located in Adventdalen (78.2°N 15.87°E) operated by the University Center in Svalbard, which measures air temperature, incoming short- and longwave radiation, relative humidity, and wind speed. Precipitation measurements are retrieved from the long-term weather station at Longyearbyen airport (9 km west of the Adventdalen weather station; 78.24°N 15.51°E) operated by the Norwegian Meteorological Institute. Precipitation is retrieved as daily values representing daily cumulative rain- or snowfall. Air temperature, relative humidity, and wind speed are measured in one-second intervals, radiation in five-minute intervals, and represent instantaneous values. The time period of measurements used in this study is 2013 to 2019 and measurements are aggregated into daily sums or averages. To create the forcing dataset, mean values of each variable for every day of the year (day-of-year average) between 2013 and 2019 are calculated to obtain a representation of current average weather conditions. Further data processing involves the classification of precipitation as rain if mean daily air temperatures are above 0°C, and as snow if air

temperatures are below 0°C. An adjustment for precipitation undercatch in Svalbard has been suggested to be 1.85 for snow and 1.15 for rain (Førland and Hanssen-Bauer, 2000), and therefore precipitation is multiplied by these respective factors. This results in an average annual sum of 330 mm for the period 2013–2019. The annual sums of rain (160 mm) and snow (170 mm w.e.) are then redistributed to equal daily amounts during the rain- and snow period, respectively. The mean annual air temperature for the calculated averages over this time period is -2.8°C. Thereby, the resulting forcing data set consists of daily values based on the average for each day of the year between 2013 and 2019 for wind speed, air temperature, incoming shortwave radiation, relative humidity, incoming longwave radiation, rain precipitation, and snow precipitation (Fig. S1). This yearly cycle of average weather data is then repeated 100 times (corresponding to 100 annual cycles) to create the forcing dataset needed to initialize and run the simulations, as described in Section 2.2.2.

2. As it currently reads, it is unclear that you did not use the averaged (spinup) data to run the simulations, which was likely the cause of some of the comments from reviewer 3.

Response: Thank you for the feedback. We made an effort to clarify in section 2.2.2 how we conduct the spinup and that the last year of the spin-up runs is used for the analysis of the results.

Changes: L211-219: In the (final) fourth step, the resulting state from the 1D single column spin-up model is mapped to each of the 33 columns of the hillslope transect model. Thereafter, the same forcing dataset (Section 2.1) is used again to run the simulations, now in the full domain allowing for all lateral and vertical dynamic processes to occur. The full model is run for 100 annual cycles, corresponding to 100 years of simulation. The first 99 years are considered as spin-up, to obtain an annually periodic steady- state for the entire surface-subsurface hillslope system in the 2D model domain. The final year of the simulation (year 99 to year 100) is then considered as the simulation result, used for analysis in this study. Thus, it is equivalent to a representation of the hydrothermal state of the subsurface corresponding to the current 2013–2019 average weather conditions. The initialization procedure is repeated for each model case considered, to ensure effects of hillslope inclination and wetness conditions are embedded in the final simulation results.

3. This unclear description in the input data leads to further confusion in section 2.2.2 (about the spinup procedure). Specifically:
L200 be specific about which data is used
L205 be specific about which data is used and that this is part of the spinup.

Response: We have improved the description in Section 2.1 (comment 1 above) to better describe the data and data processing. Therefore, we have included a reference to Section 2.1 in this section (in Section 2.2.2). We also added a brief repetition of the key information about the forcing dataset in Section 2.2.2, and restructured the paragraph describing the last step (4th step) of the spin-up to make it clearer.

Changes: Section 2.1 and L206-210: In the third step, the forcing dataset (Section 2.1) is used to bring the thermal-hydraulic conditions of the column model into an annual steady state. The annual steady state is achieved by repeating the forcing data set for 50 annual cycles, corresponding to 50 years of simulation, after which inter-annual temperature

differences throughout the column are less than 0.01°C. This procedure is necessary to obtain a physically consistent system which can be used as initial condition for the main simulation runs.

4. L258-260: This sentence needs some revision. First, there is a grammar issue ("flat cases freezes" and "slopes freezes"). Second, what is meant by "the active layer slopes"?

Response: Changed "flat cases" to "flat case". Line 259 refers to the active layer in the slopes. Added "in the".

Changes: L269-270: While the flat case freezes uniformly, the active layer in the slopes freezes faster uphill and slower downhill, causing those temperature differences.

5. Small grammar detail: The formulation "has found to…" is found in several places in the text (e.g. L29, L155, and L170). Please change to a more grammatically accurate formulation, such as "has been found to" or "was found to".

Response: Changed irritating formulation to grammatically accurate formulation according to the comment.